# Neural Topographic Factor Analysis for fMRI Data

**Eli Sennesh** *[1,3], **Zulqarnain Khan***[2], **Yiyu Wang**[3]
**Jennifer Dy**[2], **Ajay Satpute**[3], **J. Benjamin Hutchinson**[4], **Jan-Willem van de Meent**[1]
`sennesh.e@northeastern.edu, khan.zu@ece.neu.edu, wang.yiyu@northeastern.edu`
`jdy@ece.neu.edu, a.satpute@northeastern.edu, bhutch@uoregon.edu,`
`j.vandemeent@northeastern.edu`
[1] Khoury College of Computer Sciences, Northeastern University
[2] Department of Electrical and Computer Engineering, Northeastern University
[3] Department of Psychology, Northeastern University
[4] Department of Psychology, University of Oregon

## Abstract

Neuroimaging studies produce gigabytes of spatio-temporal data for a small number of participants and stimuli. Rarely do researchers attempt to model and examine how individual participants vary from each other – a question that should be addressable even in small samples given the right statistical tools. We propose Neural Topographic Factor Analysis (NTFA), a probabilistic factor analysis model that infers embeddings for participants and stimuli. These embeddings allow us to reason about differences between participants and stimuli as signal rather than noise. We evaluate NTFA on data from an in-house pilot experiment, as well as two publicly available datasets. We demonstrate that inferring representations for participants and stimuli improves predictive generalization to unseen data when compared to previous topographic methods. We also demonstrate that the inferred latent factor representations are useful for downstream tasks such as multivoxel pattern analysis and functional connectivity.

## 1 Introduction

Analyzing functional neuroimaging studies is both a large data problem and a small data problem. A single scanning run typically comprises hundreds of full-brain scans that each consist of tens of thousands of spatial locations (known as voxels). At the same time, neuroimaging studies tend to have limited statistical power [Cremers et al., 2017]; a typical study considers a cohort of 20-50 participants undergoing tens of stimuli from ten (or fewer) stimulus categories. This poses a significant problem for the over fourteen-thousand functional neuroimaging studies that seek to address both fundamental and translational research questions in cognitive neuroscience on individual differences in functional neural activity [Elliott et al., 2020]. A largely unsolved challenge in this domain is to develop analysis methods that appropriately account for both the commonalities and variations among participants and stimuli effects, scale to tens of gigabytes of data, and reason about uncertainty.

In this paper, we develop Neural Topographic Factor Analysis (NTFA)[2], a generative model for neuroimaging data that explicitly represents variation among participants and stimuli. NTFA extends Topographic Factor Analysis (TFA) and Hierarchical Topographic Factor Analysis (HTFA) [Manning et al., 2014b, 2018]. It differs from these models in that it learns a prior that maps embeddings (i.e. vectors of features) for each participant and stimulus to a conditional distribution over spatial factors and weights, instead of imposing a single global prior. The result is a structured probabilistic model that learns a representation of each participant and each stimulus.

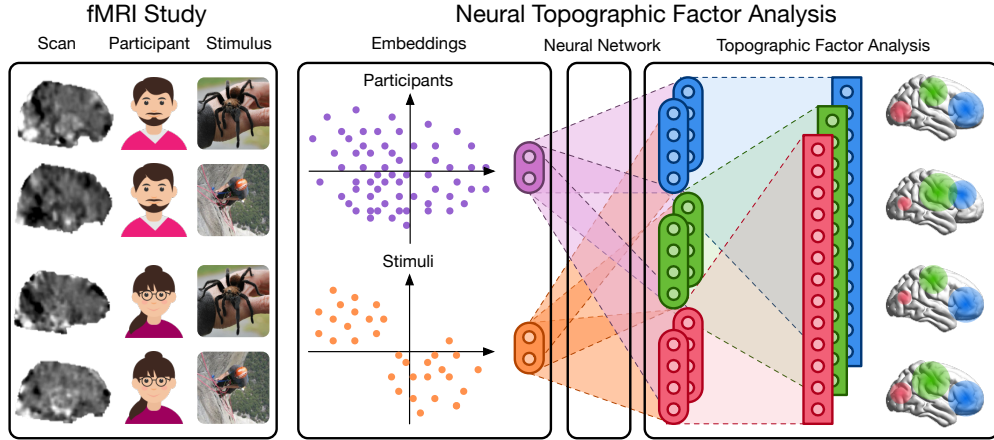

Figure 1: **Overview of Neural Topographic Factor Analysis (NTFA)**: We decompose the fMRI signal into Gaussian factors (shown in red, green and blue in the figure) that correspond to spatially and temporally related brain activity across individuals. A typical fMRI study consists of multiple trials of participants undergoing scans while experiencing different stimuli (or performing different tasks). In our generative model we represent these participants (purple) and stimuli (orange) with embedding vectors. A multilayer perceptron then predicts the factors' location, size, and weights.

NTFA offers two advantages over other dimensionality reduction methods that project data into a low-dimensional space: Our embeddings factorize the generative contributions of participants from those of stimuli, and they supply uncertainty measures by which we can measure the scale of the embedding space. Having the embedding space "scaled" by uncertainty allows us more confidence in resolving differences: if the means for embeddings of stimuli lie several standard deviations apart from each other, we can be confident they reflect significant differences in the neural repsonse.

We perform a qualitative evaluation of inferred embeddings on four datasets:

- We show that in a synthetic dataset, simulated from distinguishable clusters of participants and stimuli, inference recovers the underlying cluster structure.
- We present results for our own pilot study investigating whether threat-relevant stimuli from three categories induce the same or different patterns of neural response. NTFA infers stimulus embeddings that show differences in patterns of neural response between stimulus categories.
- We analyze and evaluate two publicly available datasets. In the first, participants with major depressive disorder and controls listened to emotionally valenced sounds and music [Lepping et al., 2016]. In the second, participants viewed images of faces, cats, five categories of man-made objects, and scrambled pictures [Haxby et al., 2001]. In both cases, NTFA infers an embedding structure that is consistent with previously reported findings.

Because NTFA is, to our knowledge, the first model to explicitly infer embeddings for participants and stimuli, we devise two simple baselines as comparisons. The first is to apply PCA directly to the input data, and the second is to compute post-hoc embeddings after training a shared response model (SRM) [Chen et al., 2015]. PCA fails to recover participant and stimulus structure, whereas the SRM yields point-estimates that are qualitatively similar but lack uncertainty estimates.

As a sanity check, we also compare predictive performance on a validation set of brain images across NTFA and HTFA. We hold out trials by their stimulus-participant pairs, requiring our model to generalize from other trials in which the same stimulus or participant were seen. PCA, the SRM, and TFA cannot recombine representations to predict such novel combinations in this way.

This work makes both neuroscientific and machine learning contributions. From a machine learning perspective, NTFA is a novel neural extension of probabilistic factor analysis methods. The inferred embeddings capture similarities in the neural response across participants and stimuli. This improves prediction on held-out data, while requiring fewer trainable parameters. From a neuroscientific perspective, the generative model in NTFA contributes to our ability to characterize individual variation in whole-brain analyses. Psychological states (e.g. emotions and memories) involve patterns of activation distributed widely throughout voxel space [Haxby et al., 2001, Satpute and Lindquist, 2019]. Existing whole-brain analyses such as multivoxel pattern analysis (MVPA) thus often rely on

Table 1: Comparison of factor analysis methods for fMRI data. When a method considers participant and stimulus variations dependently, we consider it to model variation in the independent factor. Our method (NTFA) is shown in the bottom row.

| Model | Spatial factors | Participant variation | Stimulus variation | Scanning run variation |
|-------|:---------------:|:---------------------:|:------------------:|:----------------------:|
| PCA | ✗ | ✗ | ✗ | ✗ |
| SRM | ✗ | ✓ | ✗ | ✓ |
| TFA | ✓ | ✗ | ✗ | ✗ |
| HTFA | ✓ | ✗ | ✗ | ✓ |
| TLSA | ✓ | ✓ | ✗ | ✗ |
| NTFA | ✓ | ✓ | ✓ | ✓ |

supervised feature selection using labels for stimulus categories or participant groups[Pereira et al., 2009]. In contrast, latent factors from NTFA and HTFA enable unsupervised whole-brain MVPA, and can be used to create data-driven functional connectomes.

Figure 1 outlines our proposed approach. Section 2 covers related work in factor analysis for neuroimaging data, primarily the spatially topographic methods on which we build. Section 3 develops the NTFA model. Section 4 discusses our architectural details, preprocessing steps, and experiments, then discusses and evaluates experimental results. Section 5 concludes.

## 2 Background

Factor analysis methods are widely used to reduce the dimensionality of neuroimaging data. These methods decompose the fMRI signal for a trial $Y \in \mathbb{R}^{T \times V}$ with $T$ time points and $V$ voxels into a product $Y \simeq WF$ between a lower-rank matrix of weights $W \in \mathbb{R}^{T \times K}$ and a lower-rank matrix of factors $F \in \mathbb{R}^{K \times V}$. The dimension $K \ll V$ is chosen to balance the degree of dimensionality reduction and the reconstruction accuracy.

Standard methods that are applied primarily to fMRI data include Principal Component Analysis [Abdi and Williams, 2010] and Independent Component Analysis [Hyvärinen et al., 2001]. There are also methods that have been specifically developed for fMRI analysis. These include adaptations of dictionary learning methods for large-scale fMRI datasets [Mensch et al., 2017], hyper-alignment (HA) [Haxby et al., 2011], the shared response model (SRM) [Chen et al., 2015] and the matrix-normal shared response model (MN-SRM) [Shvartsman et al., 2018] for functional alignment, as well as topographic latent source analysis (TLSA) [Gershman et al., 2011, 2014], topographic factor analysis [Manning et al., 2014b], and hierarchical topographic factor analysis [Manning et al., 2014a, 2018] for functional connectivity. Many of these come in probabilistic varieties [Cai et al., 2020].

Topographic models define spatially smooth factors via radial basis functions. Non-topographical models - such as PCA, ICA, the SRM, the MN-SRM and dictionary learning - learn a $V \times K$ factor-voxel loading matrix[3], with no requirement of spatial smoothness. Some of these learn factor-loading weights for participants, such as the SRM, while others such as dictionary learning learn factor-loading weights for experimental conditions or stimuli. HA can be considered a special case of the SRM with $K = V$. The MN-SRM is also very similar to the SRM except it enforces a weaker Gaussianity constraint on the factor-loading weights instead of the orthonormality constraint in SRM. Table 1 compares our model's latent factorization structure to those of other models. NTFA is novel in learning independent low-dimensional embeddings for both participants and stimuli.

### 2.1 Topographic Factor Analysis

This work extends TFA and HTFA, two probabilistic models that employ radial basis functions to represent spatial factors. TFA and HTFA model data that comprises $N$ trials (i.e. continuous recordings), each of which contain $T$ time points for voxels at $V$ spatial positions. TFA approximates each trial separately $Y_n \simeq W_n F_n$ as a product between time-varying weights $W_n \in \mathbb{R}^{T \times K}$ and spatially-varying factors $F_n \in \mathbb{R}^{K \times V}$. To do so, TFA assumes that the data is noisily sampled from the inner product between the weights and factors matrices

$$Y_n \sim \mathcal{N}\left(W_n F_n, \sigma^Y\right). \tag{1}$$

TFA combines this likelihood $p(Y_n \mid W_n, F_n)$ with the prior $p(W_n, F_n)$ to define a probabilistic model $p(Y_n, W_n, F_n)$. Black-box methods are then used to approximate the posterior $p(W_n, F_n \mid Y_n)$ with a mean-field variational distribution $q_\lambda(F_n, W_n)$ on the factors $W_n$ and $F_n$.

The prior $p(W_n, F_n) = p(W_n)\, p(F_n)$ factorizes over $W_n$ and $F_n$. The prior over weights $p(W_n)$ is a hierarchical Gaussian with hyperparameters $\mu^{\mathrm{W}}_{n,k}$ and $\sigma^{\mathrm{W}}_{n,k}$,

$$W_{n,t,k} \sim \mathcal{N}(\mu^{\mathrm{W}}_{n,k}, \sigma^{\mathrm{W}}_{n,k}), \qquad \mu^{\mathrm{W}}_{n,k} \sim p(\mu^{\mathrm{W}}), \qquad \sigma^{\mathrm{W}}_{n,k} \sim p(\sigma^{\mathrm{W}}), \qquad (2)$$

To define a prior over factors $p(F_n)$, TFA employs a kernel function that ensures spatial smoothness of factor values $F_{n,k,v}$ at nearby voxel positions $x^{\mathrm{G}}_v \in \mathbb{R}^3$. This kernel function $\kappa$ is normally a radial basis function (RBF), which models each factor $k \in \{1 \dots K\}$ as a Gaussian with center at a spatial location $x^{\mathrm{F}}_{n,k} \in \mathbb{R}^3$, whose width is determined by the kernel hyper-parameters $\rho^{\mathrm{F}}_{n,k}$,

$$F_{n,k,v} = \kappa(x^{\mathrm{G}}_v, x^{\mathrm{F}}_{n,k} ; \rho^{\mathrm{F}}_{n,k}), \qquad x^{\mathrm{F}}_{n,k} \sim p(x^{\mathrm{F}}), \qquad \rho^{\mathrm{F}}_{n,k} \sim p(\rho^{\mathrm{F}}). \qquad (3)$$

Interpreting factor analysis generatively enables us to incorporate additional assumptions to capture similarities across a set of related trials. HTFA [Manning et al., 2014a, 2018], introduces variables $\bar{x}^{\mathrm{F}}_k$ and $\bar{\rho}^{\mathrm{F}}_k$ representing each factor's mean positions and widths across trials,

$$x^{\mathrm{F}}_{n,k} \sim p(x^{\mathrm{F}}_{n,k} \mid \bar{x}^{\mathrm{F}}_k), \qquad \bar{x}^{\mathrm{F}}_k \sim p(\bar{x}^{\mathrm{F}}), \qquad \rho^{\mathrm{F}}_{n,k} \sim p(\rho^{\mathrm{F}}_{n,k} \mid \bar{\rho}^{\mathrm{F}}_k), \qquad \bar{\rho}^{\mathrm{F}}_k \sim p(\bar{\rho}^{\mathrm{F}}). \qquad (4)$$

HTFA assumes that brain layouts and activations across trials vary around a shared Gaussian prior. This imposes unimodality upon the distribution of neural responses across trials.

## 3 Neural Topographic Factor Analysis

NTFA extends TFA to model variation across participants and stimuli. We assume the same factor analysis model as TFA, which approximates the fMRI signal as a linear combination of time-dependent weights and spatially varying Gaussian factors. NTFA extends TFA by inferring *embedding vectors* for individual participants and stimuli. We learn a mapping from embeddings to the parameters of the likelihood model, parameterized by a neural network. Instead of HTFA's global template, we introduce factorized latent spaces of participant and stimulus embeddings, and share the neural networks mapping embeddings to factors. For reference, a complete description of our notation can be found in Table 3 in the Appendix.

The advantage of incorporating neural networks into the generative model is that it enables us to explicitly reason about multimodal response distributions and effects that vary between individual samples. The network weights $\theta$ are shared across trials, as are the stimulus and participant embeddings $z^{\mathrm{s}}_s$ and $z^{\mathrm{P}}_p$. This allows NTFA to capture statistical regularities within a whole experiment. At the same time, the use of neural networks ensures that differences in embeddings can be mapped onto a wide range of spatial and temporal responses. Whereas the hierarchical Gaussian priors in HTFA implicitly assume that response distributions are unimodal and uncorrelated across different factors $k \in [K]$, the neural network in NTFA can model such correlations by jointly predicting all $K$ factors.

We model $N$ trials in which participants $p_n \in \{1, \dots, P\}$ undergo a set of stimuli $s_n \in \{1, \dots, S\}$ and are scanned for $T$ time points per trial. We assume that participant embeddings $\{z^{\mathrm{P}}_1, \dots, z^{\mathrm{P}}_P\}$ and stimulus embeddings $\{z^{\mathrm{s}}_1, \dots, z^{\mathrm{s}}_S\}$ are shared across trials. For simplicity, we will consider the case where both embeddings have the same dimensionality $D$ and a Gaussian prior

$$z^{\mathrm{P}}_p \sim \mathcal{N}(0, I), \qquad\qquad z^{\mathrm{s}}_s \sim \mathcal{N}(0, I). \qquad (5)$$

For each participant $p$, we define the RBF center $x^{\mathrm{F}}_p$ and log-width $\rho^{\mathrm{F}}_p$ in terms of a neural mapping

$$x^{\mathrm{F}}_p \sim \mathcal{N}(\mu^x_p, \sigma^x_p), \qquad\qquad \mu^x_p, \sigma^x_p \leftarrow \eta^{\mathrm{F},x}_\theta(z^{\mathrm{P}}_p), \qquad (6)$$

$$\rho^{\mathrm{F}}_p \sim \mathcal{N}(\mu^\rho_p, \sigma^\rho_p), \qquad\qquad \mu^\rho_p, \sigma^\rho_p \leftarrow \eta^{\mathrm{F},\rho}_\theta(z^{\mathrm{P}}_p). \qquad (7)$$

Here $\eta^{\mathrm{F}}_\theta$ is a neural network parameterized by a set of weights $\theta$, which models how variations between participants and stimuli affect the factor positions and widths in brain activations. This network outputs a $K \times 4 \times 2$ tensor, that contains a two-tuple of three-dimensional parameters for each factor center $(\mu^x_p, \sigma^x_p)$ and another two-tuple of one-dimensional parameters for each factor log-width$(\mu^\rho_p, \sigma^\rho_p)$. We use a second network $\eta^{\mathrm{W}}_\theta(z^{\mathrm{P}}_p, z^{\mathrm{s}}_s)$ to parameterize the distribution over weights $W_{n,t}$ with a $K \times 2$ tensor, given the embeddings for each trial $n$ and time point $t$ with $p = p_n, s = s_n$:

$$W_{n,t} \sim \mathcal{N}\left(\mu^{\mathrm{W}}_n, \sigma^{\mathrm{W}}_n\right), \qquad\qquad \mu^{\mathrm{W}}_n, \sigma^{\mathrm{W}}_n \leftarrow \eta^{\mathrm{W}}_\theta\left(z^{\mathrm{P}}_p, z^{\mathrm{s}}_s\right). \qquad (8)$$

**Algorithm 1** NeuralTFA Generative Model

$(p_1, \ldots, p_N)$ ▷ Participant for each trial
$(s_1, \ldots, s_N)$ ▷ Stimulus for each trial
1: **for** $p$ **in** $1, \ldots, P$ **do**
2: $\quad z_p^{\mathrm{P}} \sim \mathcal{N}(0, I)$ ▷ Equation (5)
3: **for** $s$ **in** $1, \ldots, S$ **do**
4: $\quad z_s^{\mathrm{S}} \sim \mathcal{N}(0, I)$ ▷ Equation (5)
5: **for** $n$ **in** $1, \ldots, N$ **do**
6: $\quad p, s \leftarrow p_n, s_n$
7: $\quad \left(\mu_p^x, \sigma_p^x\right), \left(\mu_p^\rho, \sigma_p^\rho\right) \leftarrow \eta_\theta^{\mathrm{F}}(z_p^{\mathrm{P}})$
8: $\quad x_p^{\mathrm{F}} \sim \mathcal{N}(\mu_p^x, \sigma_p^x)$ ▷ Equation (6)
9: $\quad \rho_p^{\mathrm{F}} \sim \mathcal{N}(\mu_p^\rho, \sigma_p^\rho)$ ▷ Equation (7)
10: $\quad \mu_n^{\mathrm{W}}, \sigma_n^{\mathrm{W}} \leftarrow \eta_\theta^{\mathrm{W}}\left(z_p^{\mathrm{P}}, z_s^{\mathrm{S}}\right)$ ▷ Equation (8)
11: $\quad$ **for** $t$ **in** $1 \ldots T$ **do**
12: $\quad\quad W_{n,t} \sim \mathcal{N}(\mu_n^{\mathrm{W}}, \sigma_n^{\mathrm{W}})$ ▷ Equation (8)
13: $\quad\quad F_p \leftarrow \kappa(x_p^{\mathrm{F}}, \rho_p^{\mathrm{F}})$
14: $\quad\quad Y_{n,t} \sim \mathcal{N}(W_{n,t} \cdot F_p, \sigma^Y)$ ▷ Equation (9)

The likelihood model is the same as that in TFA,

$$Y_{n,t} \sim \mathcal{N}\left(W_{n,t} \cdot F_p, \sigma^{\mathrm{Y}}\right), \qquad\qquad F_p \leftarrow \kappa(x_p^{\mathrm{F}}, \rho_p^{\mathrm{F}}). \qquad (9)$$

We summarize the generative model for NTFA in Algorithm 1. This model defines a joint density $p_\theta(Y, W, x^{\mathrm{F}}, \rho^{\mathrm{F}}, z^{\mathrm{P}}, z^{\mathrm{S}})$, which in turn defines a posterior $p_\theta(W, x^{\mathrm{F}}, \rho^{\mathrm{F}}, z^{\mathrm{P}}, z^{\mathrm{S}} \mid Y)$ when conditioned on $Y$. We approximate the posterior with a fully-factorized variational distribution,

$$q_\lambda(W, \rho^{\mathrm{F}}, x^{\mathrm{F}}, z^{\mathrm{P}}, z^{\mathrm{S}}) = \prod_{n,t} q_{\lambda_{n,t}^{\mathrm{W}}}(W_{n,t}) \prod_s q_{\lambda_s^{\mathrm{S}}}(z_s^{\mathrm{S}}) \prod_p q_{\lambda_p^{x^{\mathrm{F}}}}(x_p^{\mathrm{F}}) \, q_{\lambda_p^{\rho^{\mathrm{F}}}}(\rho_p^{\mathrm{F}}) \, q_{\lambda_p^{\mathrm{P}}}(z_p). \qquad (10)$$

We learn the parameters $\theta$ and $\lambda$ by maximizing the evidence lower bound (ELBO)

$$\mathcal{L}(\theta, \lambda) = \mathbb{E}_q\left[\log \frac{p_\theta(Y, W, x^{\mathrm{F}}, \rho^{\mathrm{F}}, z^{\mathrm{P}}, z^{\mathrm{S}})}{q_\lambda(W, x^{\mathrm{F}}, \rho^{\mathrm{F}}, z^{\mathrm{P}}, z^{\mathrm{S}})}\right] \leq \log p_\theta(Y).$$

We optimize this objective using black-box methods provided by Probabilistic Torch, a library for deep generative models that extends the PyTorch deep learning framework [Narayanaswamy et al., 2017]. Specifically, we maximize an importance-weighted bound [Burda et al., 2016] using a doubly-reparameterized gradient estimator [Tucker et al., 2019]. This objective provides more accurate estimates for the gradient of the log marginal likelihood.

While neural network models can have thousands or even millions of parameters, we emphasize that NTFA in fact has a *lower* number of trainable parameters than HTFA. This follows from the fact that TFA and HTFA assume fully-factorized variational distributions that have $O(NK + NTK)$ parameters for $N$ trials with $T$ time points. In NTFA, the networks $\eta^{\mathrm{F}}$ and $\eta^{\mathrm{W}}$ have $O(D(D + K))$ parameters each, whereas the variational distribution has $O(D(P + S) + PK + NTK)$ parameters.

In practice, scanning time limitations impose a trade-off between $N$ and $T$. For this reason $NTK$ does not always dominate $NK$, since often $T \propto O(10)$. We can then choose $D \propto O(1)$ and $K \propto O(100)$, and if we label constant factors $c$, the total number of parameters becomes $O(cD^2 + cDK)$, making $O(cDK)$ the dominant term. When $P \ll N$, as is usually the case, NTFA can therefore have orders of magnitude fewer parameters than HTFA for $D = 2$.

## 4 Evaluation

### 4.1 Datasets

We consider four datasets in our experiments. First, we create a simulated dataset to verify that NTFA can recover a ground-truth structure in data that, by construction, contains clearly distinguishable participant and stimulus clusters (labelled "Synthetic"). Second, we analyze a previously unpublished data from a pilot study, conducted by one of the authors, that measures the neural response to threat-relevant stimuli (labelled "ThreatVids"). Third, we analyze a publicly available dataset on valenced

sounds and music, with participants divided into a control group and a depressed group [Lepping et al., 2016] (labelled "Lepping"). Finally, we verify that NTFA can reconstruct a popular publicly available dataset of participants watching pictures of animate and inanimate objects [Haxby et al., 2001] (labelled "Haxby"). These experimental datasets vary in their number of participants, time points, voxels, and task variables. A detailed description of each of these datasets can be found in Appendix A.5, and our standard neuroimaging preprocessing pipeline is discussed in Appendix A.6.

## 4.2 Model Architecture and Training

We employ participant and stimulus embeddings with $D = 2$ in all experiments. For the synthetic dataset, we analyze the data with the same number of factors as were used to generate it, $K = 3$. For non-simulated data we use $K = 100$ factors. This is somewhat fewer than previously reported for HTFA ($K = 700$) [Manning et al., 2018] owing to GPU memory limitations. We report parameter counts for HTFA and NTFA in Table 2, and provide details on network architectures in Appendix A.7.

## 4.3 Generalization to Held-out Images

To evaluate generalization, we split datasets into training and test sets, ensuring the training set contains at least one trial for each participant $p \in \{1, \ldots, P\}$ and each stimulus $s \in \{1, \ldots, S\}$. To do so, we construct a matrix of $(p, s) \in \{1, \ldots, P\} \times \{1, \ldots, S\}$ with participants as rows and stimuli as columns. We then choose all trials along the matrix's diagonals $\{n : p_n \bmod S = s_n\}$ as our test set. All other trials are used as the training set.

We evaluate generalization to held-out data in terms of the posterior-predictive probability

$$p_\theta(\tilde{Y} \mid Y) = \int p_\theta(\tilde{Y} \mid z^{\mathrm{P}}, z^{\mathrm{S}}) \, p_\theta(z^{\mathrm{P}}, z^{\mathrm{S}} \mid Y) \, dz^{\mathrm{P}} \, dz^{\mathrm{S}}.$$

Like the marginal likelihood, this quantity is intractable. We approximate it by computing a VAE-style lower bound $\mathbb{E}[\tilde{\mathcal{L}}] \leq \log p_\theta(\tilde{Y} \mid Y)$ from $L$ samples (see Appendix A.4 for a derivation),

$$\tilde{\mathcal{L}} = \frac{1}{L} \sum_{l=1}^{L} \log p\big(\tilde{Y} \mid \tilde{W}^{(l)}, \tilde{x}^{\mathrm{F}\,(l)}, \tilde{\rho}^{\mathrm{F}\,(l)}, z^{\mathrm{P}\,(l)}, z^{\mathrm{S}\,(l)}\big). \tag{11}$$

We sample embeddings from the variational distribution and remaining variables from the prior

$$z^{\mathrm{P}\,(l)} \sim q(z^{\mathrm{P}}), \qquad z^{\mathrm{S}\,(l)} \sim q(z^{\mathrm{S}}), \qquad \tilde{W}^{(l)}, \tilde{x}^{\mathrm{F}\,(l)}, \tilde{\rho}^{\mathrm{F}\,(l)} \sim p_\theta\big(\tilde{W}, \tilde{x}^{\mathrm{F}}, \tilde{\rho}^{\mathrm{F}} \mid z^{\mathrm{P}\,(l)}, z^{\mathrm{S}\,(l)}\big).$$

In Table 2, we compare NTFA to HTFA in terms of log predictive probability for held-out data, computing an importance-weighted bound over each dataset's test set. Across datasets, NTFA exhibits a higher log-likelihood and log-predictive probability than HTFA, with the same number ($K = 100$) of latent factors. We observe larger improvements by NTFA over HTFA in datasets such as ThreatVids, in which NTFA shares statistical strength across trials since $N \gg P$ and $N \gg S$.

We visualize the posterior-predictive means for held-out trials in Appendix A.3. HTFA shares its predictive distribution across all trials, because it lacks any explicit representation of participants and stimuli. By contrast, NTFA's predictive representation more closely resembles the actual data.

## 4.4 Inferred Embeddings

NTFA infers embeddings for both participants and stimuli, along with estimates of uncertainty for those embeddings. Participant embeddings appeared to primarily reflect idiosyncratic differences

Table 2: **Generalization performance** (in log predictive probability) and **parameter counts**. We approximate the log predictive with a VAE-style lower bound. We evaluate on a test set of held out subject-stimuli pairs, and use $K = 100$ factors across datasets and models. For NTFA we set $D = 2$.

| | Log-predictive HTFA | Log-predictive NTFA | Parameter count HTFA | Parameter count NTFA |
|---|---|---|---|---|
| Synthetic ($K = 3$) | $-4.72 \times 10^6$ | $\mathbf{-4.68 \times 10^6}$ | $2.16 \times 10^4$ | $1.90 \times 10^4$ |
| ThreatVids | $-2.23 \times 10^9$ | $\mathbf{-2.19 \times 10^9}$ | $1.64 \times 10^8$ | $8.88 \times 10^6$ |
| Lepping | $-2.54 \times 10^9$ | $\mathbf{-2.47 \times 10^9}$ | $2.53 \times 10^7$ | $2.61 \times 10^6$ |
| Haxby | $-7.17 \times 10^8$ | $\mathbf{-7.10 \times 10^8}$ | $1.44 \times 10^6$ | $1.01 \times 10^6$ |

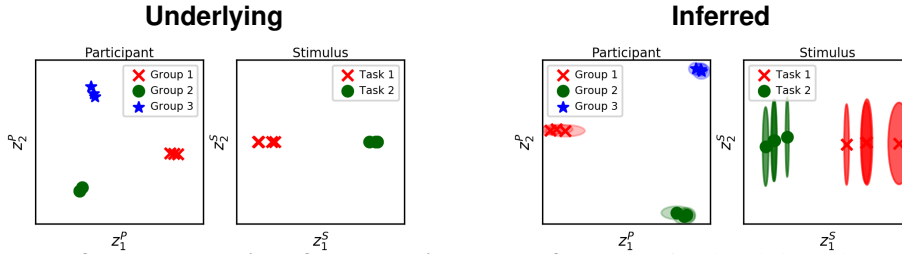

Figure 2: **Inferred embeddings for synthetic data**: **Left:** In the simulated data, three groups of participants exhibit varying levels of response in three different brain regions to *Task 1* and *Task 2* stimuli, depending on the locations of underlying participant and stimulus embeddings used to generate the data. **Right:** NTFA recovers these conditions in participant and stimulus embeddings without prior knowledge. Only the relative spatial arrangement is of interest. Since the original embeddings vary relative to each other only along the horizontal axis, NTFA learns a distribution for these embeddings with very high variance along the vertical axis.

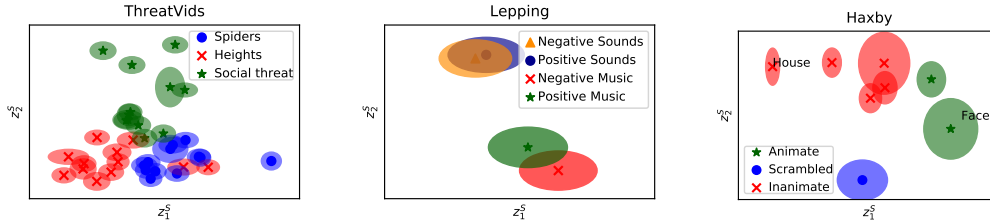

Figure 3: **Inferred distributions of stimulus embeddings. Left**: On ThreatVids, stimulus embeddings recovered groups of fear stimuli. **Middle**: On the Lepping dataset, stimulus embeddings clearly distinguish the musical stimuli from the non-musical sounds; positive vs negative music show less overlap than positive vs negative sounds. **Right**: On the Haxby dataset, stimulus embeddings for animate objects are separated from various inanimate objects. Faces were among the animate embeddings and clearly distinct from the embedding for houses, as expected [Haxby et al., 2001].

among participants, without mapping clearly to any participant conditions or behavior available in the datasets. We discuss these participant embeddings in detail in Appendix A.1. Here we discuss the extent to which stimulus embeddings align with with experimenter-defined categories.

**Synthetic Data**: For synthetic data, NTFA recovers stimulus and participant embeddings that are qualitatively similar to the embeddings that we used to generate the data (Figure 2). We emphasize that embeddings are learned directly from the synthetic data in an entirely unsupervised manner, which means that there is in principle no reason to expect embeddings to be exactly the same. However, we do observe that learned embeddings for participants and stimuli are well-separated, appear to have some variance, and are invariant under linear transformations. Moreover, given the "true" number of factors ($K = 3$), NTFA reconstructs synthetic data better than HTFA.

**ThreadVids Dataset**: In this dataset, metadata provided individual stimulus labels as well as stimulus categories to group them together. Table 2 shows that NTFA generalizes better than HTFA to held-out participant-stimulus pairs, without additional inference. In an analysis without resting-state data, NTFA uncovers stimulus embeddings that clearly correlate with stimulus categories. While "Heights" and "Spiders" show some overlap, "Social Threat" is clearly separated (Figure 3, left column).

**Lepping Dataset**: We here infer one embedding per stimulus category, since the metadata does not label individual stimuli. The embeddings (Figure 3, middle column) display a clear separation between music and nonmusical sounds, with positive and negative music showing a greater probability of differing from one-another. Positive and negative sounds overlap in the embedding space. This is consistent with previous findings [Lepping et al., 2016].

**Haxby Dataset**: We here again infer category embeddings. Inference resulted in stimulus embeddings spread throughout the embedding space, with animate and inanimate stimuli segregated (Figure 3, right column). This reflects the evidence for distinct processing of animate and inanimate objects in scenes [Naselaris et al., 2012, Blumenthal et al., 2018].

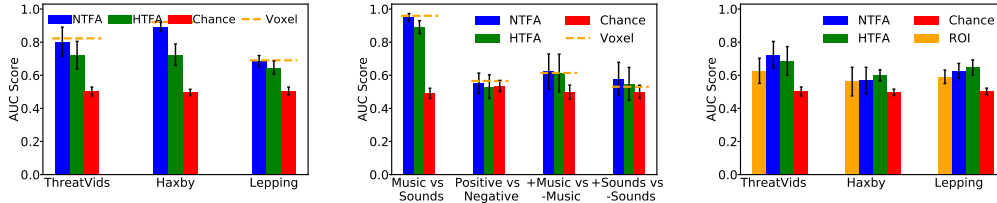

Figure 4: **Classification performance** measured by Accuracy Under the Curve (AUC). We show mean AUC scores with $95\%$ CI across categories. **Left:** For each dataset, we compare supervised voxel selection ("Voxel"), NTFA and HTFA. **Middle:** AUC scores for Lepping dataset across different stimulus categories. The embeddings in Figure 3 (middle) qualitatively match these results. **Right:** Classification using pairwise time-correlation matrices. Functional connectomes derived from NTFA and HTFA's representations outperform those from the data-agnostic regions of interest (ROIs).

## 4.5 Point-estimate Embeddings from the SRM, the MN-SRM and PCA

Evaluating the results above poses inherent challenges in the sense that we lack ground truth. Moreover, NTFA is, to our knowledge the first method that infers low-dimensional embeddings for participants and stimuli directly from the data. To provide some point of comparison, we devise two ad-hoc baselines that compute point estimates of embeddings directly from the input data.

The first baseline applies PCA, to see how a simple model might still capture meaningful structure in embeddings. We vectorized each trial $Y_n$ to obtain $N$ vectors of $TV$ dimensions. We then time-averaged these vectors, performed PCA upon them, and retained the first two principal components. This linear projection of the data did not capture any meaningful structure, as shown in Figure 9 in Appendix A.2.

The second baseline computes post-hoc embeddings from the SRM. The SRM learns a shared response matrix $S \in \mathbb{R}^{T \times K}$ and a participant-specific orthonormal weight matrix $W_p \in \mathbb{R}^{K \times V}$ to approximate the signal as $S \cdot W_p$. Since our datasets comprise unaligned stimuli, we reorder blocks in each scanning run to align stimuli across participants. We then compute participant embeddings by vectorizing the $W_p$ and projecting to the first two principal components with PCA. We split the shared-response matrix $S$ into stimulus blocks $S_s$ and then project to the first two principal components. We show results for this analysis in Figure 7 and Figure 10 in Appendix A.2.

The third baseline computes post-hoc embeddings from the Matrix Normal SRM in exactly the same fashion as done for SRM. The MN-SRM is similar to the SRM, though it assumes a weaker Gaussian prior for $W_p$ with a shared spatial covariance across subjects. It also assumes that all subjects share the same temporal noise covariance in addition to the shared response $S$. We show results for this analysis in Figure 8 in Appendix A.2.

The SRM- and MN-SRM-derived point estimates are qualitatively similar to those obtained with NTFA, but do not provide any notion of uncertainty. This makes them difficult to interpret, particularly in cases with few stimulus categories such as the Lepping and Haxby datasets.

## 4.6 Multivoxel Pattern and Functional Connectivity Analysis

One of the advantages of learning a deep generative model is that we can use the learned latent representations in downstream tasks. To illustrate this use case, we consider two types of post-scan analyses that are commonly performed on full fMRI data. As features in these analyses we use the low-dimensional representation learned by NTFA: the inferred factor locations, widths, and weights.

**Multivoxel Pattern Analysis (MVPA)**: In MVPA, a regularized linear classifier is trained to predict experimental variables from distributed patterns of mean voxel intensities. This is usually preceded by a supervised feature selection step to select voxels most relevant to the classification task [Pereira et al., 2009]. We apply this standard method to our datasets and compare it to using time-averaged weight matrices derived without supervision from NTFA and HTFA. We show the resulting classification accuracy scores, measured using Area Under the (receiver operating) Curve (AUC) on the left in Figure 4. While all three methods perform significantly better than chance, NTFA outperforms HTFA, and performs almost as well as supervised voxel selection. We also note that the stimulus embeddings qualitatively predict classification performance on different stimulus categories, as seen in the middle

of Figure 4 for Lepping dataset. NTFA learns a latent representation useful for MVPA stimulus classification, without supervision. We detail the methods and results in Appendix A.9.

**Functional Connectivity (FC)**: Functional connectivity analyses study the co-activation of brain areas during resting-state or during a task, regardless of their apparent physical distance. A variety of studies have shown FC, and changes in FC, to correlate with behavior [Elliott et al., 2019]. Voxels, however, capture neither single neurons, nor functional brain regions that could hypothetically share an activation pattern. NTFA's latent factor representations provide a data-driven alternative to standard regions of interest (ROIs) that maintains the spatial locality crucial to functional connectivity. In Figure 4 (right), we see that linear classifiers trained on NTFA's latent factor representations perform better at a stimulus classification task than those trained on ROIs. NTFA-derived FC patterns perform comparably to HTFA-derived patterns, despite NTFA's lower parameter count.

# 5 Conclusion

We have introduced Neural Topographic Factor Analysis, an unsupervised model for fMRI data that characterizes individual variation in the neural response by inferring low-dimensional embeddings for participants and stimuli. NTFA is a first step in a line of approaches that employ deep generative models to incorporate inductive biases into unsupervised analyses of neuroimaging experiments. By designing models whose structure reflects a particular experimental design, or potentially even a neuroscientific hypothesis, we can hope to appropriately account for the uncertainties that arise from limitations in statistical power and sample sizes. This provides a path towards analyses that reason about individual variation in a manner that is data-efficient and mitigates risks of overfitting the data.

# Broader Impact

While this paper reports on NTFA in terms of its characteristics as a general-purpose machine learning method for the analysis of neuroimaging data, we envision downstream impacts in the context of specific neuroscientific research questions. There is a need in neuroscience research to develop formal computational approaches that capture individual differences in neural function.

The embedding space yields a simple, visualizable model to inspect individual differences that has the potential to, at least in a qualitative manner, provide insights into fundamental questions in cognitive neuroscience. One such question is whether neural responses to stimuli are shared across individuals, vary by pre-defined participants groups (e.g. depressed vs. non-depressed participants), or are unique to participants or subgroups (e.g. as suggested by calls for "precision medicine" approaches).

Going forward, we will use our pilot data to address whether the neural basis of fear, for example, is shared across individuals and situations (i.e. there is a single "biomarker" or "neural signature" for fear), or as we expect, whether it varies by person or situation (suggesting that biomarkers for fear are idiographic) [Satpute and Lindquist, 2019]. With further developments, we plan to perform more extensive neuroimaging experiments that probe individual variation in additional fMRI datasets including in house datasets and publicly available datasets. Our hope is that the work presented in this paper will form a basis for developing probabilistic factor-analysis models with structured priors that will allow testing and development of specific neuroscientific hypotheses regarding individual variation in the functional neural organization of psychological processes.

# Acknowledgments and Disclosure of Funding

The authors thank the anonymous reviewers for their constructive feedback. We also thank Jeremy Manning for insightful conversations, and Michael Shvartsman for sharing his Matrix-Normal SRM code with us. We thank Sarah Ostadabbas and Amirreza Farnoosh for their integral involvement in the development of this project. This work was supported by startup funds from Northeastern University and the University of Oregon, as well as the Intel Corporation, the National Science Foundation (NCS 1835309), and the US Army Research Institute for the Behavioral and Social Sciences (ARI W911N-16-1-0191).

## Footnotes

[2]Source code submitted with paper and available upon request.

[3]In some cases for each subject, requiring a total of $VKP$ parameters to be learned for a study with $P$ participants, or each stimulus, requiring $VKS$ parameters, where $K$ is the number of factors and $S$ is the number of unique stimuli.

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
