[Supplementary Material]

# A Appendix

Table 3: **Description of Notations.** This table explains notations used in the paper, in the order they appear in the main text.

| Symbol | Description |
|---|---|
| $T$ | Number of TRs (in a block). |
| $V$ | Number of voxels in a brain image. |
| $K$ | Number of factors used to approximate the input data using factor analysis (usually $K << V$). |
| $Y_n \in \mathbb{R}^{T \times V}$ | $n^{\text{th}}$ block of the dataset under analysis, organized as a number of TRs times number of voxels matrix. |
| $W_n \in \mathbb{R}^{T \times K}$ | A lower-rank matrix of weights that specifies the time varying weights of each factor. |
| $F_n \in \mathbb{R}^{K \times V}$ | A lower-rank matrix of factors, such that an element $f_{kv}$ at row $k$ and column $v$ specifies the contribution of $k^{\text{th}}$ factor to the activation of voxel $v$. |
| $\sigma^Y$ | Gaussian noise variance, assumed constant across a given dataset. |
| $\mu_{n,k}^{\text{W}}, \sigma_{n,k}^{\text{W}}$ | Mean and variance for $k^{\text{th}}$ row of $W_n$. |
| $\mu^{\text{W}}, \sigma^{\text{W}}$ | Hyperparameters for $\mu_{n,k}^{\text{W}}, \sigma_{n,k}^{\text{W}}$. |
| $x_v^{\text{G}} \in \mathbb{R}^3$ | Coordinates for voxel $v$. |
| $x_{n,k}^{\text{F}} \in \mathbb{R}^3, \rho_{n,k}^{\text{F}}$ | center and log-width of the $k^{\text{th}}$ factor for block $n$. |
| $\kappa(x_v^{\text{G}}, x_{n,k}^{\text{F}} ; \rho_{n,k}^{\text{F}})$ | Radial basis function with center at $x_{n,k}^{\text{F}}$ and log-width $\rho_{n,k}^{\text{F}}$ evaluated at location $x_v^{\text{G}}$. |
| $x^{\text{F}}, \rho^{\text{F}}$ | Hyperparameters for factor centers and factor log-widths. |
| $P$ | Total number of participants in a given dataset. |
| $S$ | Total number of unique stimuli in a given dataset. |
| $p_n \in \{1, \ldots, P\}$ | Participant involved in block $n$. |
| $s_n \in \{1, \ldots, S\}$ | Stimulus involved in block $n$. |
| $z_p^{\text{P}} \in \mathbb{R}^D$ | $D-$dimensional participant embedding associated with a specific participant $p$. |
| $z_s^{\text{S}} \in \mathbb{R}^D$ | $D-$dimensional stimulus embedding associated with a specific stimulus $s$. |
| $x_p^{\text{F}} \in \mathbb{R}^{K \times 3}, \rho_p^{\text{F}} \in \mathbb{R}^K$ | Factor centers and log-widths for a specific participant $p$. |
| $\mu_p^x \in \mathbb{R}^{K \times 3}, \sigma_p^x \in \mathbb{R}^K$ | Means and variances for the $K$ factor centers for participant $p$. |
| $\mu_p^\rho \in \mathbb{R}^{K \times 3}, \sigma_p^\rho \in \mathbb{R}^K$ | Means and variances for the $K$ factor log-widths for participant $p$. |
| $\eta_\theta^{\text{F}}$ | Neural network that takes $z_p^{\text{P}}$ as input and outputs $\mu_p^x, \sigma_p^x, \mu_p^\rho, \sigma_p^\rho$ for participant $p$. |
| $\mu_n^{\text{W}} \in \mathbb{R}^K, \in \mathbb{R}^K$ | Mean and variance for each of the $K$ rows of the matrix of weights $W_n$ for block $n$. |
| $\eta_\theta^{\text{W}}$ | Neural network that takes a concatenation of $z_p^{\text{P}}$ and $z_s^{\text{S}}$ for participant $p$ and stimulus $s$ in block $n$ and outputs $\mu_n^{\text{W}}$ and $\sigma_n^{\text{W}}$. |
| $\theta$ | Learnable parameters in the generative model, that is, the neural network weights. |
| $\lambda$ | Learnable parameters of the variational distributions. |
| $\lambda_{n,t}^{\text{W}}$ | Parameters of posterior distribution over weights $W_{n,t}$. |
| $\lambda_s^{\text{S}}, \lambda_p^{\text{P}}$ | Parameters of posterior distribution over stimulus embedding $z_s^{\text{S}}$ for stimulus $s$ and participant embedding $z_p^{\text{P}}$ for participant $p$. |
| $\lambda^{x_p^{\text{F}}}, \lambda^{\rho_p^{\text{F}}}$ | Parameters of posterior distribution over factor centers $x_p^{\text{F}}$ and factor log-widths $\rho_p^{\text{F}}$ for participant $p$. |
| $\tilde{Y}$ | Held-out validation data. |

Figure 5: Participant embeddings from the ThreatVids dataset. Crosses indicate the location of the (approximate) posterior mean, and ellipses display (approximate) posterior covariance. The labels are used only for visualization purposes. Participant embeddings are color-coded by their reported level of fear across all stimulus categories (heights, spiders, and social threats). Cooler colors indicate lower mean fear ratings, while warmer colors indicate higher mean fear ratings.

## A.1 Participant embedding results

In the ThreatVids dataset, the participant embeddings uncovered three groups: the more frightened, the less frightened, and those sensitive to particular fears (Figure 5). Participant embeddings for individual fear categories are shown in Figure 5. Participants were not recruited in specific groups (e.g. arachnophobes and acrophobes), and stimuli could be categorized multiple ways (e.g. by kind or degree). We observe that most participants carried a greater fear of heights (left) and social threat (right) than of spiders (middle). A scattering of individuals in the mid-left of the embedding space appeared to suffer little overall fear in any stimulus category, while those further out from the centroid had more varied fear experiences across categories. Few individuals showed high mean fear ratings across stimulus categories.

The participant embeddings do not seem to predict the self-reported fear ratings. However as shown in Figure 5 they do seem to uncover variations among participants in the latent space. Note, for example the participant groups breaking away from the central "cluster" towards top-right and bottom-left. This suggests that there are factors not explained by the self-reported fear ratings that might be driving the individual variation in response among participants.

## A.2 "Embeddings" from PCA, SRM and MN-SRM

To establish a rough baseline. We performed PCA on the input data. The data $Y_n$ from each trial $n$ for each dataset was vectorized and these $N$ ($T \times V$ dimensional) vectors were projected to the first two principal components. Figure 10(left) shows the result overlaid with labels for each trial. Figure 9 shows the same for the three real datasets with task labels overlaid. PCA, perhaps unsurprisingly fails to capture any meaningful structure.

We also acquired a notion of post-hoc "embeddings" from SRM and MN-SRM by following these with PCA. SRM and MN-SRM learn a single shared response matrix $S \in \mathbb{R}^{K \times T}$ for all participants in an experiment and are ideally suited to experiments where the stimuli are time aligned across participants. We mimic this structure in our datasets by artificially aligning the trials in the same order of stimuli for all participants. Then the shared response matrix was split into matrices corresponding to each stimulus. These matrices were then vectorized and PCA was done on these as mentioned before. Similarly participant embeddings can be obtained by vectorizing the participant dependent weights learned by SRM and projecting them using PCA. Figure 10(right), and Figure 10(middle) show the results of this procedure for SRM and MN-SRM on synthetic data. Similarly Figure 7 and Figure 8 show the stimulus embeddings for the three real world dataset, using SRM and MN-SRM respectively. While this procedure seems to capture reasonable embeddings for the simulated data and "ThreatVids". We notice that a lack of uncertainty around these point estimates means it becomes

Figure 6: Participant embeddings for the Lepping dataset [Lepping et al., 2016]. Crosses indicate the location of the posterior mean, and ellipses display posterior covariance. The labels are used only for visualization purposes. Participant embeddings did not show a clear difference between control and major depressive groups, but did appear scattered around a linear trend in the latent space, which we have yet to interpret.

Figure 7: **Post-hoc embeddings from SRM :** Stimulus embeddings recovered post-hoc from SRM. The embeddings look qualititavely similar to NTFA, but lack uncertainty quantification which makes it difficult to meaningfully reason about the distances in the space.

Figure 8: **Post-hoc embeddings from MN-SRM :** Stimulus embeddings recovered post-hoc from MN-SRM. The embeddings look qualititavely similar to NTFA (and SRM), but lack uncertainty quantification which makes it difficult to meaningfully reason about the distances in the space.

difficult to interpret them, specially in situations where there's only a limited number of unique stimuli. As is the case for Lepping and Haxby in the middle and right of Figure 7 and Figure 8.

## A.3   Test-set predictions

To visualize the predictive distribution, we compare the time-average of $\tilde{Y}$ to a prediction $\bar{Y} = \bar{\mu}^{\mathrm{W}} \cdot \kappa(\bar{\mu}_p^x, \bar{\mu}_p^\rho)$, where $\bar{\mu}^{\mathrm{W}}, \bar{\mu}^{\mathrm{W}}, \bar{x}_p^{\mathrm{F}}, \bar{\rho}_p^{\mathrm{F}}$ are computed from the expected values $\bar{z}^{\mathrm{P}}$ and $\bar{z}^{\mathrm{S}}$ of the embeddings in the variational distribution.

Here we show test predictions from both NTFA and HTFA for our real datasets, ThreadVids (Figure 11), Lepping (Figure 12) and Haxby (Figure 13). HTFA's pale experiment-wide averages on the Lepping dataset show its inability to capture the participant- and stimulus-wise variations captured

Figure 9: **PCA projections of input data as embeddings** The baseline embeddings recovered directly from input data by projecting each trial to the first two principal components. The embeddings don't seem to capture any meaningful structure.

Figure 10: **Embeddings for synthetic data using PCA, SRM and MN-SRM**: On simulated data PCA fails to capture the participant or stimulus groups accurately. Post-hoc embeddings from the SRM and MN-SRM look qualitatively similar to NTFA albeit without uncertainty estimates.

clearly by NTFA. While HTFA does make clear predictions for its average across the Haxby dataset, nonetheless, it does not capture the variation across trials that NTFA does.

## A.4 Derivation of the lower bound to the log posterior predictive distribution

We begin by showing how to use the variational distribution to approximate the posterior predictive distribution via importance sampling, and then convert the resulting importance weight into a lower bound on the log posterior predictive. Posterior sampling from the NTFA generative model, conditioned upon the posterior distribution over embeddings, would yield the joint distribution

$$p\big(\tilde{Y}, \tilde{W}, \tilde{x}^{\mathrm{F}}, \tilde{\rho}^{\mathrm{F}}, z^{\mathrm{P}}, z^{\mathrm{S}} \mid Y\big) = p\big(\tilde{Y}, \tilde{W}, \tilde{x}^{\mathrm{F}}, \tilde{\rho}^{\mathrm{F}} \mid z^{\mathrm{P}}, z^{\mathrm{S}}\big) p\big(z^{\mathrm{P}}, z^{\mathrm{S}} \mid Y\big),$$

which factorizes according to the generative model as,

$$p\big(\tilde{Y}, \tilde{W}, \tilde{x}^{\mathrm{F}}, \tilde{\rho}^{\mathrm{F}}, z^{\mathrm{P}}, z^{\mathrm{S}} \mid Y\big) = p\big(\tilde{Y} \mid \tilde{W}, \tilde{x}^{\mathrm{F}}, \tilde{\rho}^{\mathrm{F}}, z^{\mathrm{P}}, z^{\mathrm{S}}\big) p\big(\tilde{W} \mid z^{\mathrm{P}}, z^{\mathrm{S}}\big) p\big(\tilde{x}^{\mathrm{F}}, \tilde{\rho}^{\mathrm{F}} \mid z^{\mathrm{P}}\big) p\big(z^{\mathrm{P}}, z^{\mathrm{S}} \mid Y\big).$$

The marginal of this joint distribution, that being the posterior predictive distribution, can be defined by importance weighting, where the learned variational distributions $q(z^{\mathrm{P}})$, $q(z^{\mathrm{S}})$ serve as proposals for $z^{\mathrm{P}}$ and $z^{\mathrm{S}}$ while the generative model serves as its own proposal for the other latent variables, yielding

$$p\big(\tilde{Y} \mid Y\big) = \mathbb{E}_{q,p}\left[\frac{p\left(\tilde{Y}, \tilde{W}, \tilde{x}^{\mathrm{F}}, \tilde{\rho}^{\mathrm{F}}, z^{\mathrm{P}}, z^{\mathrm{S}} \mid Y\right)}{p\left(\tilde{W}, \tilde{x}^{\mathrm{F}}, \tilde{\rho}^{\mathrm{F}} \mid z^{\mathrm{P}}, z^{\mathrm{S}}\right) q\left(z^{\mathrm{P}}\right) q\left(z^{\mathrm{S}}\right)}\right]$$

$$= \mathbb{E}_{q,p}\left[\frac{p\left(\tilde{Y} \mid \tilde{W}, \tilde{x}^{\mathrm{F}}, \tilde{\rho}^{\mathrm{F}}, z^{\mathrm{P}}, z^{\mathrm{S}}\right) p\left(\tilde{W} \mid z^{\mathrm{P}}, z^{\mathrm{S}}\right) p\left(\tilde{x}^{\mathrm{F}}, \tilde{\rho}^{\mathrm{F}} \mid z^{\mathrm{P}}\right) p\left(z^{\mathrm{P}}, z^{\mathrm{S}} \mid Y\right)}{p\left(\tilde{W} \mid z^{\mathrm{P}}, z^{\mathrm{S}}\right) p\left(\tilde{x}^{\mathrm{F}}, \tilde{\rho}^{\mathrm{F}} \mid z^{\mathrm{P}}\right) q\left(z^{\mathrm{P}}\right) q\left(z^{\mathrm{S}}\right)}\right]$$

$$= \mathbb{E}_{q,p}\left[\frac{p\left(\tilde{Y} \mid \tilde{W}, \tilde{x}^{\mathrm{F}}, \tilde{\rho}^{\mathrm{F}}, z^{\mathrm{P}}, z^{\mathrm{S}}\right) p\left(z^{\mathrm{P}}, z^{\mathrm{S}} \mid Y\right)}{q\left(z^{\mathrm{P}}\right) q\left(z^{\mathrm{S}}\right)}\right].$$

We can then apply Jensen's inequality to define a lower bound

$$
\mathrm{ELBO}_{\tilde{Y}|Y}
$$
$$
= \mathbb{E}_{q,p}\left[\log \frac{p\left(\tilde{Y} \mid \tilde{W}, \tilde{x}^{\mathrm{F}}, \tilde{\rho}^{\mathrm{F}}, z^{\mathrm{P}}, z^{\mathrm{S}}\right) p\left(z^{\mathrm{P}}, z^{\mathrm{S}} \mid Y\right)}{q\left(z^{\mathrm{P}}\right) q\left(z^{\mathrm{S}}\right)}\right]
$$
$$
\leq \log p\left(\tilde{Y} \mid Y\right).
$$

This is a standard definition of the ELBO, albeit for the posterior predictive distribution rather than the marginal likelihood (i.e. the prior predictive). By converting the log of a product of densities into a sum of log-density terms and noting that the expectations are over proposal distributions $p = p\left(\tilde{W}, \tilde{x}^{\mathrm{F}}, \tilde{\rho}^{\mathrm{F}} \mid z^{\mathrm{P}}, z^{\mathrm{S}}\right)$ and $q = q\left(z^{\mathrm{P}}\right) q\left(z^{\mathrm{S}}\right)$, we can write this ELBO as:

$$
\mathrm{ELBO}_{\tilde{Y}|Y} = \mathbb{E}_{q,p}\left[\log p\left(\tilde{Y} \mid \tilde{W}, \tilde{x}^{\mathrm{F}}, \tilde{\rho}^{\mathrm{F}}, z^{\mathrm{P}}, z^{\mathrm{S}}\right) - \log \frac{q(z^{\mathrm{P}})q(z^{\mathrm{S}})}{p(z^{\mathrm{P}}, z^{\mathrm{S}} \mid Y)}\right]
$$
$$
\mathrm{ELBO}_{\tilde{Y}|Y} = \mathbb{E}_{q,p}\left[\log p\left(\tilde{Y} \mid \tilde{W}, \tilde{x}^{\mathrm{F}}, \tilde{\rho}^{\mathrm{F}}, z^{\mathrm{P}}, z^{\mathrm{S}}\right)\right] - \mathrm{KL}\left(q\left(z^{\mathrm{P}}\right) q\left(z^{\mathrm{S}}\right) \| p\left(z^{\mathrm{P}}, z^{\mathrm{S}} \mid Y\right)\right),
$$

From the standard decomposition of the ELBO we can also reason that,

$$
\mathrm{ELBO}_{\tilde{Y}|Y} = \log p(\tilde{Y} \mid Y) - \mathrm{KL}\left(q\left(z^{\mathrm{P}}\right) q\left(z^{\mathrm{S}}\right) \| p\left(z^{\mathrm{P}}, z^{\mathrm{S}} \mid \tilde{Y}, Y\right)\right),
$$

and therefore

$$
\mathbb{E}_{q,p}\left[\log p\left(\tilde{Y} \mid \tilde{W}, \tilde{x}^{\mathrm{F}}, \tilde{\rho}^{\mathrm{F}}, z^{\mathrm{P}}, z^{\mathrm{S}}\right)\right] - \mathrm{KL}\left(q\left(z^{\mathrm{P}}\right) q\left(z^{\mathrm{S}}\right) \| p\left(z^{\mathrm{P}}, z^{\mathrm{S}} \mid Y\right)\right) =
$$
$$
\log p(\tilde{Y} \mid Y) - \mathrm{KL}\left(q\left(z^{\mathrm{P}}\right) q\left(z^{\mathrm{S}}\right) \| p\left(z^{\mathrm{P}}, z^{\mathrm{S}} \mid \tilde{Y}, Y\right)\right),
$$
$$
\mathbb{E}_{q,p}\left[\log p\left(\tilde{Y} \mid \tilde{W}, \tilde{x}^{\mathrm{F}}, \tilde{\rho}^{\mathrm{F}}, z^{\mathrm{P}}, z^{\mathrm{S}}\right)\right] = \log p(\tilde{Y} \mid Y)
$$
$$
- \mathrm{KL}\left(q\left(z^{\mathrm{P}}\right) q\left(z^{\mathrm{S}}\right) \| p\left(z^{\mathrm{P}}, z^{\mathrm{S}} \mid \tilde{Y}, Y\right)\right)
$$
$$
+ \mathrm{KL}\left(q\left(z^{\mathrm{P}}\right) q\left(z^{\mathrm{S}}\right) \| p\left(z^{\mathrm{P}}, z^{\mathrm{S}} \mid Y\right)\right).
$$

Since the variational distributions $q$ were already optimized during training to minimize the KL divergence in the third term on the right hand side of the above equation, we can reason that it will be small compared to the KL divergence in the second term (between the variational distribution and the true posterior given the test data). The difference of KL's should therefore remain nonnegative, allowing us to use the expected log-likelihood as a lower bound to the log posterior predictive probability of the test data. Additionally, the low dimensionality ($D = 2$ in our experiments) of $z^{\mathrm{P}}$ and $z^{\mathrm{S}}$ compared to $Y$ led the log likelihood to dominate the ELBO in all our experiments, a fact which should not be changed by passing to $ELBO_{\tilde{Y}|Y}$. This leads to Equation (11) in Section 4.

### A.5 Description of datasets

**Threat Videos ("ThreatVids")**[4]: A fundamental question in affective neuroscience is whether threat-relevant stimuli from different categories involve a single or multiple distinct systems [Wager et al., 2015]. To evaluate whether NTFA can provide insight into this fundamental research question, we conducted and analyzed our own study. 21 participants each watched 36 videos depicting threat-related content involving spiders, looming heights, and social evaluative threat (12 videos per category). Each video was approximately 20 seconds long and was followed by a set of self-report ratings and a rest period. The videos were chosen to vary in how much fear they normatively evoke within each category. This data contains 81,638 voxels (white matter removed) and 552 time points per scanning run for three runs. Using NTFA, we examine whether neural activity justified organization of the stimuli into three categories.

Figure 11: **Predictive distribution for the ThreatVids dataset**: We compare time-averaged held-out data (*left*) to the posterior-predictive mean for three trials. In NTFA (*center*), the learned generative model and inferred embeddings inform the distribution for unseen participant-stimulus combinations. In HTFA (*right*) the predictive distribution is the same for trials, since the shared global template in this model does not differentiate between participants and stimuli.

Figure 12: **Test predictions for the Lepping dataset**: We show average images for three trials, with participant-stimulus pairs held out from the training set. Posterior predictive estimates under NTFA (center) capture meaningful trial-specific variation in the original images (left), whereas HTFA can only re-use its same global template across differing trials (right).

Figure 13: **Test predictions for the Haxby dataset**: We show average images for three trials, with participant-stimulus pairs held out from the training set. Posterior predictive estimates under NTFA (center) capture meaningful trial-specific variation in the original images (left), whereas HTFA can only re-use its same global template across differing trials (right).

**Emotional Musical and Nonmusical Stimuli in Depression ("Lepping")** [Lepping et al., 2016][5]:
19 participants with major depressive disorder and 20 control participants (P=39) underwent musical
and nonmusical stimuli to examine neural processing of emotionally provocative auditory stimuli in
depression. In each trial, participants listened to music or nonmusical valenced (positive or negative)
sounds, interleaved with trials in which they heard neutral tones. The fMRI data had 62,733 voxels
(white matter removed) and 105 time points in each scanning run for five runs.

**Face and Objects Image Viewing ("Haxby")** [Haxby et al., 2001][6]: fMRI was used to measure
whole brain response while subjects viewed faces, cats, five categories of man-made objects, and
scrambled pictures. The study consisted of six subjects (P=6) and 12 scanning runs per subject, with
32,233 voxels (white matter removed) and 121 time points for each scanning run.

Table 4: **Dataset Summary**

|           | No. Participants P | No. of Stimuli S | TRs per block T | No. of Voxels V |
|-----------|--------------------|------------------|-----------------|-----------------|
| Synthetic | 9                  | 8                | 20              | 5,000           |
| ThreatVids| 23                 | 36               | 20              | 81,638          |
| Lepping   | 39                 | 4                | 10              | 62,733          |
| Haxby     | 6                  | 8                | 12              | 32,233          |

## A.6  Preprocessing

The raw BOLD signal collected in fMRI is generally not usable for analysis. It contains both
physiological (cerebrospinal fluids, global signal, and white matter) and motion artifacts. We employ
standard neuroimaging preprocessing - including slice timing correction, high pass filtering and
spatial smoothing - for all fMRI data using fMRIPrep [Esteban et al., 2019]. The processed data
still has units that are incomparable across scanning runs. For the ThreatVids and Haxby datasets,
we z-scored each task trial with respect to the entire set of rest trials (trials in which no stimulus
was presented) within each run. For the Lepping dataset, we treated neutral tones as rest trials and
performed the same z-scoring procedure. This provides a common scale of units across trials within
a dataset, capturing meaningful difference in activation intensities relative to neutral conditions ("rest"
and "tones"). Since neural activity peaks about three seconds after the task onset [Aguirre et al.,
1998], we make sure to account for this delay when loading each dataset by offsetting the stimulus
onsets by three seconds. We input the resulting z-scored data to NTFA, and use it for all evaluations.

## A.7  Neural network architectures and initialization

The network $\eta_\theta^F(\cdot)$ is a multilayer perceptron (MLP) with one hidden layer and PReLU activations.
We extract the factor parameters F, $x$ and F, $\rho$ by viewing the $8K$-dimensional result as a $K \times 4 \times 2$
tensor. The network $\eta_\theta^W(\cdot)$ is similarly an MLP with one hidden layer, though operating over both
embeddings. We extract the weight parameters $w_n$ by casting its $2K$-dimensional result as a $K \times 2$
matrix. Architectural details for both networks are given in Table 5. The neural network weights $\theta$
specify the linear layers of the networks.

We train the parameters $\theta$ and $\lambda$ on all models using the Adam optimizer [Kingma and Ba, 2015]
for 1000-1500 epochs per dataset. We use one particle to calculate the IWAE-style bound to the log-
evidence and its gradient estimator at training time, with a learning rate $\eta_\lambda = 0.01$ and $\eta_\theta = 0.0001$.
We anneal the learning rate with a patience of 100 epochs, and a multiplicative decline of 0.5.

Similarly to Manning et al. [2018], we employ a K-means initialization in all experiments across
models, initializing the variational parameters for HTFA and the bias in the final layer of the generative
model for NTFA.

Table 5: **Network architectures for $\eta_\theta^{\mathrm{F}}$ and $\eta_\theta^{\mathrm{W}}$**

| Layer | $p_\theta\left(x_p^{\mathrm{F}}, \rho_p^{\mathrm{F}} \mid z_p^{\mathrm{P}}\right)$ | $p_\theta\left(W_{n,t} \mid z_p^{\mathrm{P}}, z_s^{\mathrm{S}}\right)$ |
|---|---|---|
| Input | $z_p^{\mathrm{P}} \in \mathbb{R}^D$ | $z_p^{\mathrm{P}}, z_s^{\mathrm{S}} \in \mathbb{R}^D$ |
| 1 | FC $D \times 2D$ PReLU | FC $2D \times 4D$ PReLU |
| 2 | FC $2D \times 4D$ PReLU | FC $4D \times 8D$ PReLU |
| 3 | FC $4D \times 8K$ | FC $8D \times 2K$ |
| Output | $\left(\mu_p^x, \sigma_p^x, \mu_p^\rho, \sigma_p^\rho\right) \in \mathbb{R}^{8K}$ | $\left(\mu_n^{\mathrm{W}}, \sigma_n^{\mathrm{W}}\right) \in \mathbb{R}^{2K}$ |

## A.8 Synthetic data generation

We consider a simulated dataset in which there are three participant groups (*Group 1*, *Group 2* and *Group 3*) of three participants each. All participants underwent two categories of hypothetical stimuli, called *Task 1* and *Task 2*, with four stimuli within each category. Each participant underwent one hypothetical scanning run with rest trials interleaved between stimuli. We manually defined three distinct factors in a standard MNI_152_8mm brain. We then sampled participant embeddings $\{z_1^{\mathrm{P}}, ..., z_9^{\mathrm{P}}\}$ and stimulus embeddings $\{z_1^{\mathrm{S}}, ..., z_8^{\mathrm{S}}\}$, from mixtures of three and two distinct Gaussians respectively. We set the means for these Gaussians to meet the following conditions under noisy combination. **1.** All participants show no whole-brain response during rest except random noise. **2.** Under *Task 1* stimuli, Group 1 exhibits approximately half the response in the first region as compared to under *Task 2* stimuli. The rest of the brain shows no response. Similarly, Group 2 and Group 3 exhibit a response in the second and third regions respectively, while the rest of the brain shows no response. **3.** Each stimulus in *Task 1* and *Task 2* provokes a response lower or higher than the stimulus category's average based on the stimulus embedding's location. Algorithm 2 shows the pseudocode for generating synthetic datasets similar to this synthetic data used in this paper. We will also include the exact script with the code repository for our method.

---

**Algorithm 2** Generating a simple Synthetic data to test NTFA using Nilearn [Abraham et al., 2014]. For a dataset with $T$ time points per block, $K$ factors, $C$ stimulus categories, $N_C$ stimuli per category, $G$ participant groups, and $N_G$ participants per group. This leads to $S = C * N_C + (2N_C + 1)$ stimuli (to allow for interleaved rest blocks) and $P = G * N_G$ participants.

---
1: Load Template Brain ▷ e.g. MNI_152_8mm
2: Define $\mu_{1...C}^S, \Sigma_{1...C}^S$ ▷ means and covariances for embeddings for each stimulus category.
3: Define $\mu_{1...G}^P, \Sigma_{1...G}^P$ ▷ means and covariances for each participant group.
4: $x_{1,...,K}^F \leftarrow K$ ▷ manually selected voxels.
5: Define $\sigma^x, \mu^\rho, \sigma^\rho$ ▷ variance for factor centers, means and variance for log-width
6: $\rho_{1,...,K} \sim \mathcal{N}(\mu^\rho, \sigma^\rho)$
7: $F \leftarrow \mathrm{RBF}(x, \rho)$ ▷ create factor matrix using radial basis functions
   Order the total stimuli according to required experiment design and save indices accordingly.
   e.g. Category 1, Rest, Category 2, Rest, and so on.
8: **for** $c$ **in** $1, \ldots, C$ **do**
9:    **for** $s$ **in** $1, \ldots N_C$ **do** $z_s^c \sim \mathcal{N}(\mu_c^S, \Sigma_c^S)$
   ▷ generate stimulus embeddings
10: **for** $g$ **in** $1, \ldots, G$ **do**
11:    **for** $p$ **in** $1, \ldots N_G$ **do** $z_p^g \sim \mathcal{N}(\mu_g^P, \Sigma_g^P)$
   ▷ generate participant embeddings
12: **for** $g$ **in** $1, \ldots, G$ **do**
13:    **for** $p$ **in** $1, \ldots N_G$ **do**
14:       **for** $s$ **in** $1, \ldots, S$ **do**
15:          **for** $k$ **in** $1, \ldots, K$ **do**
16:             $W_{[k,s:s+T]} \sim \mathcal{N}(0, \sigma^W)$ ▷ if $s$ is the start of a rest block
17:             $W_{[k,s:s+T]} \sim \mathcal{N}(z_p^T z_s, \sigma^W)$
   $Y_{p+(g-1)*N_G} = WF$ ▷ data for one participant
---

## A.9  MVPA Classification

In this section we provide details of the classification pipeline as well as results beyond those presented in Figure 4. The traditional pipeline outlined in Pereira et al. [2009] was used to do classification on the input data. One classical approach is to first select a subset of voxels with reliably different mean intensities between the experimental variables being tested. Usually this is done by selecting 500 voxels based on the $f$-statistic from an analysis of variance (ANOVA). After this supervised feature selection step, a linear support-vector machine (SVM) is usually trained (without hyperparameter tuning) over some combination of cross-validation scans.

For each block (an instance of a participant undergoing a stimulus), only the mean voxel activity was considered. We employed a leave-out-one cross-validation approach with respect to scanning runs for each subject, with a one-vs-all linear SVM trained and tested for each stimulus category. For Haxby and ThreatVids, we used a leave-one-out cross-validation approach on scanning runs, while for Lepping (in which the experimental design did not support leaving whole scanning sessions out) we applied a stratified three-fold cross-validation scheme across all trials with the same stimulus. We then ran the same classification pipeline again, substituting NTFA's and HTFA's MAP estimates of the weight matrix $W$ for the label-supervised voxels.

For each cross validation run, the feature selection of voxels was done by keeping the top $500$ of the voxels with the most reliable differences in the mean intensity for the stimulus category the classifier was being trained for vs the remaining categories. The linear SVM trained on these selected voxels on training runs was then tested on the held-out runs. Since this is a one-vs-all scheme, the classes are unbalanced, and raw accuracy can be inflated just by predicting the most frequent class. We therefore report Area Under the ROC Curve (AUC) instead.

For NTFA and HTFA, we use the same pipeline, except there is no supervised feature selection step. Instead, we used MAP estimates of generated $W \in \mathbb{R}^{T \times K}$ matrices, averaged across time. These were employed as the training features for classifiers with respect to the cross-validation scheme above. As is evident from Figure 4, and Tables 6 and 7, NTFA performs similarly to the supervised pipeline above, and often outperforms HTFA.

Based on suggestion from a reviewer, we also considered including NTFA training within the cross validation folds. However, we note that this is computationally very expensive, since MVPA here is done with separate cross-validation folds for each subject. This would result in training NTFA more than 50 times for the three real datasets. Moreover, given the unsupervised nature of NTFA, we believe that this shortcut of training NTFA on all data only once to extract features is unlikely to be problematic.

Table 6: **Classification Details** on ThreatVids dataset. "Voxel" indicates the ANOVA-SVM strategy on input data. NTFA performs consistently better than HTFA and closer to performing a completely supervised feature selection + classification pipeline on the input data.

| Subject | Voxel Heights | Voxel Social | Voxel Spiders | NTFA Heights | NTFA Social | NTFA Spiders | HTFA Heights | HTFA Social | HTFA Spiders |
|---|---|---|---|---|---|---|---|---|---|
| 4 | .90 ± .03 | .94 ± .04 | .96 ± .06 | **.86 ± .01** | .89 ± .09 | **1.00 ± .00** | .84 ± .07 | **.90 ± .10** | .86 ± .06 |
| 5 | .91 ± .08 | .97 ± .03 | 1.00 ± .00 | **.85 ± .03** | **.91 ± .11** | **.97 ± .03** | .58 ± .10 | .62 ± .17 | .76 ± .15 |
| 6 | .42 ± .08 | .26 ± .09 | .54 ± .08 | **.45 ± .11** | .27 ± .06 | **.61 ± .09** | .30 ± .14 | **.35 ± .13** | .58 ± .15 |
| 7 | .44 ± .03 | .38 ± .06 | .39 ± .05 | .44 ± .00 | **.44 ± .09** | **.36 ± .00** | **.59 ± .06** | .25 ± .00 | .30 ± .11 |
| 8 | .99 ± .01 | .86 ± .10 | 1.00 ± .00 | **.96 ± .01** | **.91 ± .05** | **.99 ± .01** | .79 ± .10 | .80 ± .04 | .96 ± .03 |
| 9 | .56 ± .17 | .46 ± .05 | .57 ± .15 | .52 ± .12 | .36 ± .06 | .42 ± .13 | **.74 ± .10** | **.77 ± .16** | **.64 ± .10** |
| 10 | .97 ± .04 | .96 ± .03 | 1.0 ± .00 | **.99 ± .01** | **.95 ± .05** | **1.0 ± .00** | .93 ± .04 | .94 ± .04 | .95 ± .04 |
| 11 | .86 ± .13 | .93 ± .03 | .94 ± .07 | **.85 ± .14** | **.90 ± .03** | **.86 ± .08** | .66 ± .04 | .84 ± .08 | .69 ± .17 |
| 12 | .36 ± .19 | .67 ± .06 | .65 ± .18 | **.49 ± .08** | .53 ± .10 | **.61 ± .17** | .38 ± .22 | **.60 ± .08** | .51 ± .11 |
| 13 | .88 ± .09 | .99 ± .02 | .85 ± .11 | **.91 ± .08** | **.98 ± .02** | **.86 ± .10** | .80 ± .15 | .87 ± .03 | .86 ± .03 |
| 14 | .85 ± .03 | .89 ± .05 | .97 ± .03 | .81 ± .00 | .89 ± .02 | **1.0 ± .00** | **.87 ± .03** | **.94 ± .03** | .98 ± .01 |
| 15 | .82 ± .10 | .87 ± .09 | .92 ± .08 | **.84 ± .07** | **.93 ± .05** | **.95 ± .07** | .60 ± .11 | .85 ± .08 | .83 ± .13 |
| 16 | .96 ± .03 | .99 ± .01 | .95 ± .01 | **.96 ± .03** | .99 ± .01 | .96 ± .03 | .85 ± .11 | **1.0 ± .00** | .93 ± .05 |
| 17 | 1.0 ± .00 | .92 ± .05 | .83 ± .07 | .91 ± .04 | **.83 ± .04** | **.88 ± .04** | **.95 ± .02** | .47 ± .12 | .81 ± .15 |
| 18 | .81 ± .14 | .68 ± .11 | .81 ± .08 | .63 ± .14 | .58 ± .17 | .59 ± .07 | **.68 ± .07** | **.59 ± .08** | **.70 ± .05** |
| 19 | .85 ± .01 | .94 ± .04 | .85 ± .09 | **.89 ± .03** | **.93 ± .00** | **.84 ± .08** | .81 ± .04 | .67 ± .14 | |
| 23 | .70 ± .05 | .85 ± .06 | .72 ± .07 | **.81 ± .03** | **.81 ± .03** | **.85 ± .07** | .58 ± .02 | .66 ± .09 | .75 ± .09 |
| 25 | .84 ± .06 | .95 ± .03 | .99 ± .01 | **.76 ± .04** | .93 ± .07 | **.99 ± .01** | .61 ± .12 | **.94 ± .02** | .90 ± .04 |
| 26 | .84 ± .16 | .53 ± .03 | .82 ± .10 | .64 ± .11 | **.76 ± .05** | .81 ± .09 | **.77 ± .05** | .59 ± .00 | **.95 ± .05** |
| 28 | .78 ± .07 | .97 ± .03 | 1.0 ± .00 | **.96 ± .05** | **.97 ± .00** | **1.0 ± .00** | .83 ± .15 | .87 ± .05 | .99 ± .01 |
| 29 | .95 ± .07 | .94 ± .04 | 1.0 ± .00 | .87 ± .09 | .93 ± .04 | .93 ± .05 | **.90 ± .03** | **.95 ± .01** | **.95 ± .04** |

Table 7: **Classification Details** on Lepping dataset. "Voxel" indicates the ANOVA-SVM strategy on input data. NTFA performs consistently better than HTFA and closer to performing a completely supervised feature selection + classification pipeline on the input data.

| Subject | Voxel -Music | Voxel -Sounds | Voxel +Music | Voxel +Sounds | NTFA -Music | NTFA -Sounds | NTFA +Music | NTFA +Sounds | HTFA -Music | HTFA -Sounds | HTFA +Music | HTFA +Sounds |
|---|---|---|---|---|---|---|---|---|---|---|---|---|
| CTRL-1 | .75 ± .20 | .42 ± .43 | .58 ± .24 | .79 ± .12 | .75 ± .20 | .29 ± .21 | **.96 ± .06** | .58 ± .31 | **.96 ± .06** | .37 ± .27 | .92 ± .12 | **.83 ± .24** |
| CTRL-2 | .62 ± .27 | .92 ± .12 | .92 ± .12 | .54 ± .06 | **.79 ± .21** | **1.0 ± .00** | .79 ± .16 | .75 ± .20 | .67 ± .31 | .92 ± .12 | .71 ± .21 | .58 ± .24 |
| CTRL-3 | .12 ± .10 | .54 ± .29 | .71 ± .21 | .29 ± .21 | .58 ± .12 | .79 ± .29 | .87 ± .10 | .58 ± .12 | .71 ± .21 | .83 ± .12 | .79 ± .212 | .42 ± .12 |
| CTRL-4 | .79 ± .16 | .79 ± .16 | .67 ± .12 | .67 ± .12 | **.92 ± .12** | **.62 ± .10** | .75 ± .20 | .67 ± .12 | .87 ± .18 | .58 ± .12 | **.83 ± .12** | .58 ± .12 |
| CTRL-5 | .71 ± .25 | .54 ± .33 | .83 ± .24 | .62 ± .10 | .67 ± .12 | **.67 ± .31** | **.87 ± .12** | .62 ± .18 | .75 ± .20 | .67 ± .24 | .83 ± .20 | **.83 ± .12** |
| CTRL-6 | .50 ± .41 | .79 ± .16 | .67 ± .24 | .83 ± .24 | .42 ± .24 | **.83 ± .12** | .92 ± .12 | **1.0 ± .00** | .58 ± .42 | .75 ± .20 | **.96 ± .06** | **1.0 ± .00** |
| CTRL-7 | .79 ± .16 | .42 ± .42 | .58 ± .12 | .50 ± .20 | **.92 ± .12** | .42 ± .42 | .33 ± .12 | .25 ± .20 | .83 ± .12 | .46 ± .41 | **.50 ± .20** | .17 ± .24 |
| CTRL-8 | 1.0 ± .00 | .83 ± .12 | .86 ± .10 | .75 ± .20 | **1.0 ± .00** | .75 ± .00 | **1.0 ± .00** | **1.0 ± .00** | .67 ± .12 | **.92 ± .12** | **1.0 ± .00** | .92 ± .12 |
| CTRL-9 | .58 ± .12 | .96 ± .06 | .71 ± .21 | .50 ± .00 | .42 ± .31 | **.92 ± .12** | **.92 ± .12** | .67 ± .12 | .75 ± .00 | **.92 ± .12** | .87 ± .10 | .58 ± .24 |
| CTRL-10 | .62 ± .31 | .87 ± .10 | .37 ± .10 | .17 ± .12 | .37 ± .44 | **.83 ± .12** | .25 ± .20 | .21 ± .16 | **.92 ± .12** | **.83 ± .12** | .75 ± .20 | **.50 ± .20** |
| CTRL-11 | .46 ± .21 | .87 ± .18 | .54 ± .16 | 1.0 ± .00 | .33 ± .12 | **1.0 ± .00** | .90 ± .12 | **1.0 ± .00** | .58 ± .12 | .79 ± .16 | **.50 ± .20** | **1.0 ± .00** |
| CTRL-12 | .83 ± .12 | .67 ± .24 | .87 ± .10 | .92 ± .12 | **.75 ± .20** | **.67 ± .31** | **.83 ± .12** | .71 ± .06 | .75 ± .20 | .58 ± .12 | .79 ± .83 | **.83 ± .24** |
| CTRL-13 | .71 ± .26 | .67 ± .31 | .92 ± .12 | .29 ± .26 | **.92 ± .12** | **.75 ± .35** | **.75 ± .20** | .29 ± .26 | .67 ± .24 | .25 ± .20 | .71 ± .21 | **.75 ± .20** |
| CTRL-14 | .58 ± .31 | .67 ± .31 | .33 ± .12 | .83 ± .12 | .62 ± .31 | .33 ± .12 | .83 ± .12 | .75 ± .20 | .67 ± .24 | .46 ± .16 | **.42 ± .12** | **.87 ± .18** |
| CTRL-15 | .62 ± .18 | .87 ± .18 | .58 ± .31 | .92 ± .12 | .58 ± .24 | .67 ± .47 | .50 ± .12 | .62 ± .10 | .58 ± .12 | **.79 ± .29** | .08 ± .12 | **.87 ± .18** |
| CTRL-16 | .87 ± .18 | .79 ± .21 | .46 ± .16 | .58 ± .31 | **.92 ± .12** | .67 ± .12 | .33 ± .31 | .50 ± .20 | .87 ± .10 | **.83 ± .12** | **.50 ± .41** | .42 ± .24 |
| CTRL-17 | .67 ± .31 | .25 ± .20 | .79 ± .16 | .62 ± .10 | .37 ± .10 | .21 ± .21 | **.83 ± .12** | .62 ± .10 | **.71 ± .26** | **.29 ± .26** | **.79 ± .16** | **.79 ± .21** |
| CTRL-18 | .71 ± .21 | .62 ± .37 | .83 ± .12 | .83 ± .12 | .83 ± .24 | .67 ± .12 | **.92 ± .12** | .62 ± .17 | **.92 ± .12** | **1.0 ± .00** | **.92 ± .12** | .46 ± .33 |
| CTRL-19 | .87 ± .10 | 1.0 ± .00 | .54 ± .36 | .92 ± .12 | **.96 ± .06** | .75 ± .202 | .50 ± .41 | .92 ± .12 | .83 ± .24 | **.96 ± .06** | .33 ± .24 | .58 ± .24 |
| CTRL-20 | .83 ± .24 | .83 ± .12 | .54 ± .33 | .50 ± .00 | **.79 ± .16** | .42 ± .42 | .42 ± .42 | .37 ± .10 | .58 ± .31 | .67 ± .12 | .46 ± .39 | **.62 ± .31** |
| MDD-1 | .54 ± .36 | .75 ± .20 | .83 ± .12 | .08 ± .12 | **.79 ± .16** | .75 ± .20 | .833 ± .12 | .17 ± .12 | .67 ± .12 | .67 ± .31 | **1.0 ± .00** | .17 ± .12 |
| MDD-2 | .67 ± .24 | .25 ± .00 | .83 ± .24 | .67 ± .31 | **.67 ± .31** | .17 ± .12 | .75 ± .20 | **.67 ± .31** | **.67 ± .47** | .42 ± .24 | **.79 ± .29** | **.71 ± .06** |
| MDD-3 | .11 ± .16 | 1.0 ± .00 | .58 ± .31 | .25 ± .35 | **.44 ± .34** | **.89 ± .16** | .56 ± .42 | .83 ± .12 | .19 ± .14 | **.89 ± .16** | .64 ± .31 | .00 ± .00 |
| MDD-4 | .79 ± .16 | .62 ± .10 | .75 ± .20 | .83 ± .12 | **.83 ± .12** | .58 ± .31 | .75 ± .20 | .75 ± .00 | .71 ± .06 | .67 ± .24 | .54 ± .33 | **.87 ± .10** |
| MDD-6 | .67 ± .31 | .58 ± .42 | .87 ± .10 | .92 ± .12 | .67 ± .31 | **.75 ± .35** | .92 ± .12 | .62 ± .31 | .71 ± .21 | .62 ± .18 | **.96 ± .06** | **.79 ± .21** |
| MDD-7 | .54 ± .16 | .79 ± .16 | .42 ± .31 | .58 ± .31 | .54 ± .21 | **.83 ± .12** | **.75 ± .00** | .62 ± .10 | .58 ± .12 | .71 ± .06 | .42 ± .31 | |
| MDD-8 | .83 ± .12 | .58 ± .31 | .79 ± .21 | .42 ± .31 | **.92 ± .19** | .42 ± .12 | .71 ± .33 | .33 ± .471 | .87 ± .10 | .54 ± .26 | **.83 ± .12** | .33 ± .47 |
| MDD-9 | 1.0 ± .00 | .67 ± .12 | .87 ± .18 | .96 ± .06 | **1.0 ± .00** | .25 ± .20 | **1.0 ± .00** | **.92 ± .12** | **1.0 ± .00** | .29 ± .33 | .75 ± .20 | .83 ± .12 |
| MDD-10 | 1.0 ± .00 | .87 ± .18 | .83 ± .24 | .58 ± .12 | **1.0 ± .00** | **.92 ± .12** | **.83 ± .24** | .50 ± .00 | .92 ± .12 | .87 ± .18 | .79 ± .21 | **.71 ± .21** |
| MDD-11 | .62 ± .10 | .67 ± .12 | .79 ± .21 | .83 ± .12 | .54 ± .16 | **.83 ± .12** | .87 ± .18 | .79 ± .16 | .46 ± .06 | .37 ± .31 | **1.0 ± .00** | .75 ± .20 |
| MDD-12 | .83 ± .24 | .42 ± .12 | .58 ± .12 | .79 ± .16 | **.71 ± .33** | **.33 ± .12** | **.71 ± .21** | .58 ± .12 | .46 ± .16 | .17 ± .12 | .67 ± .12 | .37 ± .10 |
| MDD-13 | .67 ± .31 | .71 ± .26 | .25 ± .00 | .58 ± .12 | .92 ± .12 | **1.0 ± .00** | .42 ± .12 | .58 ± .12 | **1.0 ± .00** | **.92 ± .12** | .58 ± .12 | **.75 ± .20** |
| MDD-14 | .58 ± .24 | .83 ± .12 | .58 ± .31 | .58 ± .12 | .33 ± .12 | **.79 ± .16** | .50 ± .20 | .67 ± .31 | .37 ± .10 | .62 ± .10 | **.67 ± .12** | .58 ± .24 |
| MDD-15 | 1.0 ± .00 | .58 ± .24 | .75 ± .20 | .58 ± .12 | **.83 ± .12** | **.75 ± .20** | **.83 ± .24** | .75 ± .20 | .58 ± .31 | .50 ± .41 | .79 ± .21 | .67 ± .12 |
| MDD-16 | .92 ± .12 | .46 ± .39 | .29 ± .26 | .37 ± .10 | **.96 ± .06** | .46 ± .39 | **.54 ± .39** | .54 ± .06 | **.96 ± .06** | .62 ± .31 | .42 ± .24 | .37 ± .10 |
| MDD-17 | 1.0 ± .00 | .21 ± .06 | .67 ± .24 | .75 ± .20 | **1.0 ± .00** | .25 ± .20 | .50 ± .00 | .83 ± .12 | .87 ± .18 | .25 ± .00 | **.79 ± .21** | **1.0 ± .00** |
| MDD-18 | .83 ± .24 | 1.0 ± .00 | .75 ± .00 | .92 ± .12 | **.83 ± .24** | .33 ± .24 | **.92 ± .12** | **.92 ± .12** | .42 ± .12 | .29 ± .21 | .83 ± .12 | .58 ± .31 |
| MDD-19 | .75 ± .00 | .62 ± .18 | .58 ± .24 | .92 ± .12 | **.75 ± .00** | .62 ± .10 | **.75 ± .00** | **.83 ± .12** | **.92 ± .12** | **.75 ± .20** | .58 ± .12 | .58 ± .24 |

## Footnotes

[4]This dataset is currently in preparation for online repository pending deidentification and submission of an empirical report.

[5]This data was obtained from the OpenfMRI database. Its accession number is ds000171.

[6]https://openneuro.org/datasets/ds000105/