[Reviews · NeurIPS 2020]

Review 1

Summary and Contributions: This paper develops the Neural Topographic Factor Analysis (NTFA) as a new probabilistic factor analysis model that can infer neural activities profile from functional neuroimages. The proposed method can learn Spatio-temporal information in the fMRI datasets to distinguish signal rather than (white) noise. The empirical studies show that inferring representations for participants and stimuli may improve the prediction rate of a classification model on the testing phase.

Strengths: This paper wants to address an important topic for the task-based fMRI analysis. Overall this paper can be an interesting study.

Weaknesses: The proposed method is benchmarked with only three datasets — including one private data, and two public datasets. Although the empirical studies are convincing, only 3 datasets are not enough to evaluate the quality of a new machine learning approach.

Correctness: 1. In this paper, the notations are confusing. In the regular papers, scalers are denoted by small letters, vectors are defined with small letters (highlighted by bold), and matrices are denoted by capital letters using bold. Presenting a notation table in Supplementary Material can also be useful. 2. The abstract can present a better problem definition for general audiences. The proposed method may have some applications in other areas of machine learning. 3. Dataset information can be presented in the form of a table in Supplementary Material — i.e., number of subjects, number of time points, number of voxels, TR, TE, the vendor of fMRI machines, etc. 4. The limitations/applications of the proposed method must be discussed in detail. 5. What is that “.” in (1), (9), and definition of Y? If it is a dot product, you have to remove it — “W_nF_n” instead of “W_n.F_n” (such as other parts of your paper).

Clarity: In this paper, two concepts are not clear: 1. SRM (functional alignment) versus TFA (connection analysis): it seems that this paper wants to show that SRM and TFA are developed for addressing the same problem — i.e., there are two general factor analysis approaches. However, the primary concerns for functional alignment and connection analysis are different. I recommend you to see the discussion presented in this paper: “Incorporating structured assumptions with probabilistic graphical models in fMRI data analysis,” Cai et al. 2020. 2. Latent factor analysis versus embeddings: Latent analysis refers to finding a lower-dimensional (vector) space — where you enable to provide a better feature representation. In general, when you embedded a (vector) space, you move it to a higher dimension — which can increase the danger of overfitting in fMRI analysis. These concepts must clearly explain in this paper. There are some cases in this paper that these concepts are used instead of each other.

Relation to Prior Work: The proposed method must be compared with “Matrix-normal models for fMRI analysis”, Shvartsman et al. 2018. While the compared techniques in Table 1 did not claim that they can provide spatial and temporal analyses at the same time, the proposed method and the matrix-normal SRM (Shvartsman et al. 2018) want to address almost the same problem.

Reproducibility: Yes

Additional Feedback: There are some minor linguistic and typo problems in this paper. UPDATE: I have read the comments on this submission, as well as the authors' response. The responses addressed my concerns. I have decided to increase my rating to “A good submission, accept".


Review 2

Summary and Contributions: This paper describes a new probabilistic factor analysis model for the analysis of fMRI data. The goal is to learn a low-dimensional representation of stimuli and participants that collectively explain the high-dimensional fMRI responses. The model builds upon the previously described TFA model, which models each fMRI trial as the product of a low-dimensional, time-varying weight matrix and low-dimensional spatially-varying factor matrix. The factor matrix is explicitly constrained to be spatially localized using an RBF kernel, thus promoting latent factors that are more similar to brain regions. The present model extends TFA by making the spatial centers and widths a nonlinear function of a low-dimensional set of participant embedding; the nonlinear function is parameterized by a neural network. Similarly centers and spreads of the time-varying weight matrix are parameterized as a nonlinear function of both the participant embeddings and low-dimensional stimulus embeddings. Inference is accomplished using a fully factorized variational scheme across the latents. The authors report improvements over NTFA (another hierarchical extension of TFA): better log-predictive performance on held-out stimulus/participant and better stimulus decoding accuracy. They also show that the stimulus embeddings uncovered by the model reflect meaningful structure in the stimuli. They do not find any obvious or meaningful structure in the participant embeddings.

Strengths: The model is well-motivated and the implementation appears solid. It is natural to want to have a way of jointly inferring participant and stimulus embeddings that collectively explain high-dimensional fMRI data, since such a model could be broadly useful for both basic and clinical research.

Weaknesses: I found the model comparisons unsatisfying. There are no quantitative comparisons with methods other than NTFA, and the quantitative comparisons that are shown lack stats. The input to the model is structured as a 3D tensor (time x trial x voxel), and so for the stimulus analysis, it would be natural to compare with tensor decomposition methods (e.g. CP, tucker). The PCA baseline is a straw-man, particularly because they are vectorizing across time and voxels. It would be better to aggregate responses across time (e.g. average with some delay) and then perform PCA. -- The authors have addressed all of these points in the response -- I would guess that the vast majority of the cross-subject variation is due to noise, since noise by definition is unique to a participant and there is plenty of noise in any fMRI recording. I'm thus unsurprised that the participant embedding do not appear meaningful. I suspect that to make progress on discovering meaningful participant-specific factors, the model needs some way of distinguishing reliable cross-subject variation from unreliable cross-subject variation. This is a major limitation in my opinion because the participant embedding are one of the major innovations here. It's not obvious to me that the stimulus embeddings buy you that much compared with alternative models. -- This issue was not addressed by the authors. To me, the participant embeddings are the main new thing, but it is not clear whether there is any hope of this method being able to distinguish meaningful variation from noise (e.g. due to motion). --

Correctness: There are no statistics. -- Response indicates this will be addressed -- It would be helpful to have a measure of prediction accuracy with more meaningful units than log-predictive accuracy. For example, perhaps they could measure and report the correlation in left-out test data. As is it's hard to know whether the performance boosts are substantial or not. -- Not addressed -- It would be nice to know how important the nonlinear mapping is to the model. Would the results be qualitatively different using a linear mapping? -- Addressed thoroughly in author response --

Clarity: I overall enjoyed reading the paper. There is very little detail describing the MVPA or functional connectivity analyses, which makes them hard to evaluate, and impossible to replicate. -- Not addressed, probably due to limited space limitations --

Relation to Prior Work: Yes

Reproducibility: No

Additional Feedback:


Review 3

Summary and Contributions: The authors present a factor analysis technique to model patient and stimuli from fMRI data. Their method extends previous work as it allows to capture patient variations as well as stimulus variations using a generative approach. They display that the obtained embeddings reflect patient groups for simulated data and stimulus categories for real data. They further show that the obtained embeddings can be used in downstream tasks such as classification.

Strengths: The method proposed seems reasonable and provides good results, especially when assessing the embeddings of the stimuli. Being able to learn embeddings from fMRI data would be an important result, especially given the amount of small datasets available.

Weaknesses: I would have enjoyed more in depth analyses of the parameters like D and K, especially on the simulated data. Some analyses could be improved (as detailed below).

Correctness: I found areas of improvement but did not identify major flaws in this work that would prevent me from trusting the presented results. Detailed comments: - For the simulated data, have the authors investigated the effect of D and K? It is a good first step to ensure that, given the correct parameters, the method can recover the ground truth. In real-world applications, this ground truth is absent. Therefore, it would be interesting to see how many "false positives" or "false negatives" in terms of latent dimensions the approach is leading to. How would this affect downstream tasks? - number of parameters. This analysis is a bit convoluted and depends on arbitrary choices such as D and K. In addition, P is not necessarily orders of magnitude smaller than N, but will be compared to NxT (as mentioned by the authors above, this is a trade-off). Can there be a simpler way to rephrase saying that for specific choices of D and K, the proposed method can have a smaller number of parameters? I would also add that the number of parameters is not the only factor in a model's complexity, so would suggest not emphasizing this too much. The mention of limited GPU compared to previous work is a bit of a contradictory signal. - In most MVPA approaches, the goal is to predict on new subjects given previous subjects. The held-out strategy includes all subjects in the training set. Is this a limitation of the method that it can't be used on new subjects? - "NTFA's predictive representation more closely resembles the actual data". Was there any quantification of this? - PCA: I would have expected an averaging across T to increase the signal-to-noise ratio. Is there a reason why temporal scans were concatenated instead? - SRM baseline: Is the reordeing not affecting the temporal correlation patterns of the signals? In this case, wouldn't the data be considered as "more noisy" and more difficult to fit? (more below)

Clarity: The paper is well written and notation is clear. I have minor comments: - There is not enough context to assess the correctness of Table 1. It might be better located in Appendix with enough descriptions of previous work. - It is unclear why stimulus variation is a desirable variation factor to isolate. The design of neuroimaging studies is thorough and typically defines "categories" of stimuli. The interest then lies is investigating the response to one category compared to another. As the authors mention, variations across stimuli from one category is considered as noise. What goal would be served by looking at this signal? If the idea is to discard the categories, then this should be clearer in the text as this could have important ramifications in the future. - If an analysis is detailed in the main text, its results should also be detailed and illustrated in the main manuscript. I understand there are space constrains, but it might be better to focus on a couple of results and show them fully (e.g. I would have liked to see the HTFA results on the synthetic data) and leave other analyses to Appendix. - Appendix references in the main text do not correspond to the mentioned sections. - Please provide an honest analysis of the method, including discussing its limitations. - The functional connectivity seems to perform classification as an end-task. This is a bit confusing wrt to the title of the section. Do you mind revising those titles? It seems to me that the downstream task is similar across both experiments.

Relation to Prior Work: Yes, although baselines for some analyses were not detailed enough in the main text and did not seem ideal (see comments).

Reproducibility: Yes

Additional Feedback: Further detailed feedback (character count limits): - MVPA analysis: it is a false belief that feature extraction should be performed before analysis. If not performed correctly (e.g. by not appropriately splitting the dataset), these can hurt performance. Why not use a whole-brain analysis based on kernel methods? Linear kernel SVMs have shown high performance (e.g. on the Haxby dataset) and multiple toolboxes can be used for this analysis. Hyper-parameter tuning is also recommended. - While Pereira et al., 2009 is a good reference, it is a bit outdated with regards to whole-brain pattern recognition analyses. For example, leave-one-out cross-validation has been to shown to display overfitting and using 10 to 20% of the data as validation set is typically recommended. I understand this is not the main result of the paper, but would advise the authors to use a commonly used toolbox in the domain (many exist in Python or Matlab) for these analyses. - From my understanding of the supplementary materials, feature selection was performed within the cross-validation (as it should), but NTFA was performed outside. What explains this discrepancy? This is would then lead to a bias in the comparison (i.e. using all data for embedding but only training data for feature extraction). - Studying the effect of the size of the embedding used on downstream tasks would be interesting: could this be a hyper-parameter to optimize? - For functional connectivity, the difference between methods does not seem significant. Please be more cautious in your wording. - Future work: Have the authors tried to concatenate multiple datasets?The fact that different studies have (even slightly) different designs typically prevents them to be used jointly. Given the current approach, wouldn't this be a possibility? It would be interesting to see how subjects from different datasets cluster, especially if investigating similar conditions. Reproducibilty: Code is provided with the manuscript for reproducibility. Broader Impact: This should not be an extension of the discussion/conclusion. No negative applications were mentioned. Response: I thank the authors for providing a detailed response, which refers to additional experiments and discussion. I feel these will make the paper stronger, and am interested at how much variation in embeddings will be observed by performing NTFA in the cross-validation.

[Author Response · NeurIPS 2020]

We would first like to thank the reviewers for their thoughtful commentary and constructive feedback, particularly given the constraints imposed by the COVID-19 pandemic. We are happy to see that the reviewers agree that studying individual differences is an important problem for neuroimaging, and overall appear to lean towards acceptance.

The reviewers do have a number of comments and questions relating to baselines, the importance of characterizing stimulus variation, and the number of available datasets for experiments. Many of these comments are actionable and addressable by camera ready, as we discuss below.

**R2: Comparison to MN-SRM, tensor decompositions.** We appreciate R2's helpful reference to Shvartsman et al. (2018), which appears to consider spatial variation more explicitly than the original SRM. We were not able to find an open-source implementation of the MN-SRM and as a result were not able to perform a comparison in time for this response. We will attempt to implement this method ourselves and include a comparison in the camera-ready. We will also evaluate whether CP/Tucker tensor decomposition methods could serve as additional baselines, as suggested by R3.

**R3+R5: PCA baseline.** R3 notes that PCA is not a strong baseline. We agree and it is not intended as such; we will emphasize this more clearly. We appreciate the suggestion from R3 and R5 to perform time-averaging before doing PCA to improve it as a baseline. We did so and found that this did not result in qualitatively different embeddings from the non-averaged analysis in appendix Section A.2. We will update the manuscript and the figures to reflect this change.

**R3: How important are the nonlinearities?** We have have followed up and trained a version of NTFA in which we replace dense networks with a single linear layer. On our datasets, this results in equivalent reconstruction performance, at the cost of the inferred embeddings looking meaningless (the participant embeddings collapse into the Gaussian prior, and stimulus embeddings lack the interpretable pattern seen in Figure 3).

**R5: Cross-validation in MVPA analysis.** R5 writes, "feature selection was performed within the cross-validation (as it should be), but NTFA was performed outside." We will clarify that for MVPA, we treat NTFA as an unsupervised feature extractor before any supervised training is performed. This means there is no supervised feature selection. We believe that unsupervised training on all data (which was done as a computational shortcut) is likely not problematic. To verify this, we will re-train NTFA independently for each fold and re-run our analysis.

**R2: Notation and dataset tables.** We appreciate these suggestions and will include tables that summarize notation, as well as tables with summary statistics for each dataset.

**R2: SRM vs TFA.** We agree that SRM and TFA as methods have different intended cases (functional alignment and connectivity respectively) and will explain this more clearly in the text. We will also cite Cai et al. (2020).

**R3+R5 Why examine stimulus variation?** In standard neuroimaging analysis, researchers designate categories of stimuli prior to the experiment and model the BOLD signal with a Gaussian (or similar location-scale) distribution. This approach assumes that variation among stimuli can be treated as noisy deviations from a single mode. However, the assumption that experimenter-designated categories are optimal has been challenged. Cognitive and translational neuroscience suggests that the same category of stimulus-induced state (e.g. fear) or phenotypic trait (e.g. depression), may involve multiple distinct neural pathways. Rather than assume that experimenter-designated categories are correct, NTFA enables researchers to test this assumption. If the stimuli group together into categories naturally, as we found in some of our experiments, then the categorical assumption may be considered well-justified. Where significant intra-category variance appears, NTFA enables researchers to then delve deeper into why stimuli show divergent effects and how that impacts overall conclusions from the findings.

**R2+R3+R5: Limitations of NTFA and our present experiments.** The three reviewers point out that more interesting evaluation results could be obtained by applying NTFA to further datasets, and that interpretation of participant embeddings has been left to future work. We agree that NTFA is a first step in a longer research program. Participant embeddings in the current model are difficult to interpret because they serve a dual role: they capture both the spatial alignment of latent factors within voxel-space and the per-participant variations in BOLD response to stimuli. The main factor that limits our ability to improve the current model is the availability of datasets that are suitable to the study of individual differences. We are in the process of performing a more comprehensive version of the ThreadVids pilot study, which we hope will allow us to improve upon our current results in future work. We will add discussion to this effect to the manuscript.

[Meta-Review · NeurIPS 2020]

Three knowledgeable reviewers found that the paper is a solid piece of work that tries to solve a difficult fMRI problem. The reviewers found some limitations in the paper such as the lack of statistical evaluations of the results, and the lack of comparisons, both of which can be addressed in the final version of the manuscript. The author response addressed most of the reviewer's concerns (please make sure to address the remaining ones as well).